# Emission metrics for quantifying regional climate impacts of aviation
*Marianne T. Lund[*,1], Borgar Aamaas[1], Terje Berntsen[1,2], Lisa Bock[3], Ulrike Burkhardt[3], Jan S.*
*Fuglestvedt[1], Keith P. Shine[4]*
*1 CICERO, Center for International Climate Research, Oslo, Norway*
*2 Department of Geosciences, University of Oslo, Norway*
*3 Deutsches Zentrum für Luft- und Raumfahrt, Institut für Physik der Atmosphäre,*
*Oberpfaffenhofen, Germany*
*4 Department of Meteorology, University of Reading, UK*
**ABSTRACT**
This study examines the impacts of emissions from aviation in six source regions on global and
regional temperature. We consider the NOx-induced impacts on ozone and methane, aerosols and
contrail-cirrus formation, and calculate the global and regional climate metrics Global Warming
Potential (GWP), Global Temperature change Potential (GTP) and Absolute Regional
Temperature change Potential (ARTP). GWPs and GTPs vary by a factor 2-4 between source
regions. We find the highest aviation aerosol metric values for South Asian emissions, while
contrail-cirrus metrics are higher for Europe and North America, where contrail formation is
prevalent, and South America plus Africa, where the optical depth is large once contrails form.
The ARTP illustrate important differences in the latitudinal patterns of radiative forcing (RF) and
temperature response: The temperature response in a given latitude band can be considerably
stronger than suggested by the RF in that band, also emphasizing the importance of large-scale
circulation impacts. To place our metrics in context, we quantify the temperature response in the
four broad latitude bands following a one-year pulse emission from present-day aviation, including
$CO_2$. Aviation over North America and Europe causes the largest net warming impact in all latitude
bands, reflecting the higher air traffic activity in these regions. Contrail-cirrus gives the largest
warming contribution in the short-term, but also remains important in several regions even after
100 years. $CO_2$ becomes dominant at a 100 year time horizon. However, our results also illustrate
the relative importance of $CO_2$ on shorter time scales. Our emission metrics can be further used to
estimate regional temperature impact under alternative aviation emission scenarios. A first
evaluation of the ARTP in the context of aviation suggests that further work to account for vertical
sensitivities in the relationship between RF and temperature response would be valuable for further
use of the concept.






## 1 INTRODUCTION

The global aviation sector has historically been one of most rapidly growing economic sectors, and the increase in activity is projected to continue in the foreseeable future. The impacts of aviation exhaust emissions on atmosphere and climate have been under scrutiny for several decades (e.g., (Brasseur et al., 2016; Fahey et al., 1995; Lee et al., 2009; Penner et al., 1999; Sausen et al., 2005). Today, global aviation contributes about 2% of the total anthropogenic $CO_2$ emissions. In addition to emissions of $CO_2$, aviation impacts climate through a number of other mechanisms, including emissions of nitrogen oxides (NOx), aerosols and precursor species, aerosol-cloud interactions and formation of contrail-cirrus. These have a much shorter lifetime than a perturbation to $CO_2$ and hence produce distinctly inhomogeneous radiative forcing and contribute to further inhomogeneity in temperature response. Moreover, the regional and global climate impact of equal emissions of short-lived species can vary depending on where, and even when, the emissions occur (e.g, Berntsen et al. (2006); Stevenson and Derwent (2009)). Knowledge of such regional and temporal variability is important for understanding the climate impacts of the sector and can be an important consideration in mitigation.

The spatial variability – from emissions through impacts on atmosphere and radiative forcing, to temperature response – that characterizes the aviation sector is well recognized in the scientific community. Several studies have explored the regional differences in aviation $NO_x$-induced ozone changes and quantified the radiative forcing (RF) of aviation emissions (Gilmore et al., 2013; Köhler et al., 2013; Lee et al., 2009; Olsen et al., 2013; Penner et al., 1999; Sausen et al., 2005; Stevenson & Derwent, 2009; Unger et al., 2013). However, fewer estimates of regional temperature response exist (Huszar et al., 2013; Jacobson et al., 2012; Olivié et al., 2012). Because of the lack of a one-to-one relationship between forcing and response patterns (Boer & Yu, 2003; Shindell et al., 2010), the strength of regional aviation-induced temperature changes cannot be inferred directly from the corresponding RF distributions. The only tools to provide temperature response and other climate variables on very detailed spatial scales (e.g., grid point level) are comprehensive climate or earth system models. However, most, if not all, individual economic sectors or individual mitigation measures cause small perturbations, making it difficult (or costly) to capture a robust signal of the consequent response in climate models without significantly

scaling up the emissions. On the other hand, knowledge about contributions of individual sectors
to total climate impact, and the effects of specific measures, is essential for the formulation and
assessment of effective mitigation strategies.

Emission metrics, such as the Global Warming Potential (GWP) and Global Temperature change
Potential (GTP) are tools which can serve as a bridge to policy making, and are commonly used
for aggregating information and placing different emissions on a common scale. Several studies
have also used simplified climate models to calculate the global mean temperature response to
aviation (Berntsen & Fuglestvedt, 2008; Khodayari et al., 2013; Lee et al., 2009; Marais et al.,
2008; Skeie et al., 2009). While aggregation and synthesis is often necessary for reasons of
applicability, any such spatially aggregated measure has the disadvantage that it hides the
underlying spatial distributions of impacts and the strength of regional impacts.

Recent work has advanced the development of metric concepts which can capture regional impacts.
Shine et al. (2005a) and Lund et al. (2012) explored the use of non-linear damage functions to
capture spatial information about climate impacts in  global metrics. Lund et al. (2012) compared
the impact of NOx and aerosol emissions from the transport sectors and found that the loss of
information due to global averaging was largest in the case of aviation. However, currently the
only metric to provide estimates of impacts on a sub-global scale is the Absolute Regional
Temperature change Potential (ARTP) (Shindell & Faluvegi, 2009; Shindell & Faluvegi, 2010).
The ARTP uses a set of regional climate sensitivities to provide time-varying surface temperature
response in four latitude bands to emissions, accounting for the regional RF caused by the
emissions. These sensitivities are derived from simulations with a coupled climate model and
express the relationship between the pattern of a radiative forcing and the consequent surface
temperature in a given latitude band. Hence, the ARTP provides additional insight into the
geographical distribution of temperature change beyond that available from traditional global
metrics. For instance, Collins et al. (2013) calculated  ARTPs  for emissions of short-lived climate
forcers in four regions, while Lund et al. (2014) used the ARTP to quantify regional temperature
impacts of on-road diesel emissions and Sand et al. (2016) examined Arctic temperature responses.

In this study we calculate GWP, GTP and ARTP for global and regional aviation emissions. We
consider a broad set of forcing mechanisms and emissions in six separate source regions. Aviation-
induced radiative forcing of ozone and aerosols are obtained from simulations with the chemistry-
transport model OsloCTM3 (Søvde et al., 2012) and subsequent radiative transfer calculations,
while the radiative forcing from the formation contrail-cirrus is simulated with ECHAM5-CCMod
(Bock & Burkhardt, 2016a; b). Based on this we calculate both global and regional emission
metrics. This allows us to capture *(i)* the impact of regional aviation emissions on global
temperature response and *(ii)* the regional temperature response to regional emissions. Using the
ARTP, we quantify the regional impact of the present-day (i.e., year 2006) aviation sector, showing
the contributions over time from individual species and emission regions to the temperature
response in different latitude bands. The set of regional climate sensitivities that form the basis for
the ARTP, by expressing the inter-regional relationship between radiative forcings and
temperature response, have so far only been derived by one climate model and for four broad
latitude bands (Shindell & Faluvegi, 2009). Establishing such sensitivities requires a large number
of multi-decadal simulations and is thus very costly in terms of computer resources. Taking our
analysis one step further, we therefore compare the regional temperature response to aviation
ozone and black carbon aerosols estimated using these regional climate sensitivities with results
from simulations with three other climate models, hence performing a first evaluation of the
application of the ARTP in the context of selected aviation forcing mechanisms.

**2 METHODOLOGY**
**2.1 ATMOSPHERIC CONCENTRATIONS AND RADIATIVE FORCING**
To quantify the changes in atmospheric concentrations of ozone and aerosols resulting from global
and regional aviation emissions, the global chemistry-transport model OsloCTM3 is used (Søvde
et al., 2012). The model is run with year 2010 meteorology and a T42 resolution (approximately
2.8° x 2.8°) with 60 vertical layers from the surface to 0.1 hPa. Aviation emissions for 2006 are
from the AEDT inventory (Wilkerson et al., 2010), while other anthropogenic and biomass burning
emissions are from the HTAPv2 (Janssens-Maenhout et al., 2015) and Global Fire Emissions
Database version 3 (GFED3; van der Werf et al. (2010)) inventories. Guidance on how to access
and use the AEDT emissions in atmospheric models is provided by Barrett et al. (2010). For input
to the OsloCTM3, emissions are interpolated to the model's horizontal and vertical resolution, and
averaged monthly. A 20% perturbation of aviation emissions of black and organic carbon (BC,
OC), sulfur dioxide ($SO_2$) and NOx is applied globally and in six separate emission source regions,
covering both hemispheres and the main flight corridors (Fig. 1): North America (NAM), Europe
(EUR), East Asia (EAS), South Asia and Middle East (SAS), South America and Africa (SAF)
and South Pacific Ocean (SPO). Total aviation emissions by species and region are summarized
in Table SI1.

For input to the metric calculations (Sect. 2.2), we calculate the global mean RF for each emitted
species $i$ and emission region $r$, as well as the RF averaged over four latitude bands $l$ (90-28°S,
28°S-28°N, 28-60°N and 60-90°N) ($RF_{i,l,r}$). The direct forcing of aviation aerosols is quantified
using the 3D radiative forcing kernels developed by Samset and Myhre (2011), where the radiative
forcing per burden was derived by imposing globally uniform perturbations of aerosol
concentrations at 20 different pressure levels from the surface to 20 hPa. The NOx-induced ozone
forcing is calculated using the Oslo radiative transfer model (RTM), including stratospheric
temperature adjustment. The Oslo RTM consists of a broad band scheme for longwave radiation
and a scheme using the multi-stream DISORT code for shortwave radiation (Myhre et al., 2000).
The RF of NOx-induced methane changes is calculated as
$$RF_{CH4,r} = \Delta\tau_{CH4,r} \cdot [CH4]_{2010} \cdot RFeff_{CH4} \cdot f \qquad\qquad (1)$$
where $\Delta\tau_{CH4,r}$ is the relative change in methane lifetime between the control run and each of the
emissions perturbations, $[CH4]_{2010}$ the 2010 global methane concentration of 1788 ppb, $RFeff_{CH4}$
the methane radiative efficiency of 0.36 (mW m$^{-2}$) ppb$^{-1}$ (IPCC, 2014) and $f$ the feedback factor of
1.34 (Holmes et al., 2013). The RF of the subsequent methane-induced $O_3$ change ($O_3$LL) (Wild
et al., 2001) is calculated as:
$$RF_{O3LL,r} = 0.5 \cdot RF_{CH4} \qquad\qquad (2)$$
A decrease in atmospheric methane also results in a slight decrease in stratospheric water vapor,
and hence an additional small negative impact, included in our $RF_{CH4}$ based on Myhre et al. (2007)
as:
$$RF_{SWV,r} = 0.15 \cdot RF_{CH4} \qquad\qquad (3)$$
To obtain the latitudinal distributions $RF_{CH4,l,r}$ and $RF_{O3LL,l,r}$ we use the same approach as in Collins
et al. (2013) and Lund et al. (2014), and scale results according to the latitudinal distribution of
methane forcing derived from a global methane concentration perturbation (Fry et al., 2012).
In order to quantify the RF from the formation and persistence of contrail-cirrus caused by global
and regional aviation emissions, simulations with ECHAM5-CCMod (Bock & Burkhardt, 2016a;
b) are performed at T42 resolution with 41 vertical levels using the same emission inventory and
source regions as in the OsloCTM3. ECHAM5-CCMod is based on the ECHAM5-HAM
(Lohmann et al., 2008), which is extended by a contrail cirrus scheme with two-moment
microphysics. The two-moment microphysical scheme allows for a more realistic representation
of the microphysical and optical properties of contrail cirrus. The model is validated and used to
provide updated calculations of stratosphere adjusted contrail-cirrus RF by Bock and Burkhardt
(2016b), resulting in a global mean RF of 56 mW m$^{-2}$ for the 2006 AEDT aviation emissions used
in this study. The existence of contrail-cirrus results in a decrease in natural cirrus clouds, causing
a negative RF that partly offsets contrail-cirrus warming. The magnitude of this feedback effect is
uncertain, but  estimates suggest a forcing on the order of 20 percent of the RF of contrail-cirrus
on global mean (Burkhardt & Kaercher, 2011). We include the feedback by reducing the contrail-
cirrus RF by 20 percent for all emission source regions, i.e., assuming, in the absence of more
detailed information, that the negative RF is spatially constant.

The RFs are given by component, source region and latitude band in Table SI2. Most of the species
have short atmospheric lifetimes and consequently the RF is largest in the latitude bands closest to
where the emissions occur. Some contrail-cirrus RF values are negative, which might be due to a
change of cloud cover overlap in the model. It should be noted that there is a broad range in the
estimates of  RF caused by the various aviation emissions reported in the literature (e.g., Brasseur
et al. (2016); Lee et al. (2009)) and such uncertainties in RF will propagate to the emissions metrics.
To examine the importance, we perform an uncertainty analysis as described in Sect. 2.2. Moreover,
our results do not include effects of aerosol-cloud interactions, which is an important caveat.
Studies have suggested a potential impact of aviation BC on large scale cirrus clouds, but have yet
to agree even on the sign of the radiative forcing (Zhou & Penner, 2014). A few studies have
investigated effects of aviation emissions on liquid clouds, with global mean RF estimates ranging
from -46 to -15 mW/m$^2$ (Gettelman & Chen, 2013; Kapadia et al., 2016; Righi et al., 2016), i.e., a
negative RF that could offset a considerable fraction of the positive RF of contrail-cirrus and ozone
on a global scale. However, at present uncertainties in these estimates are also very large, and we
consider that their inclusion here would be premature.

## 208 2.2 GLOBAL AND REGIONAL CLIMATE METRICS

We present calculations of the global and regional climate metrics GTP, GWP and ARTP for
regional aviation emissions. The GWP and GTP methodology is extensively documented in the
literature, (e.g., Aamaas et al. (2013); Fuglestvedt et al. (2003); Shine et al. (2005b)). Hence, we
only describe the ARTP framework here.
The Absolute Regional Temperature change Potential (ARTP) gives the time-dependent
temperature response following a pulse emission in the four latitude bands 90-28°S, 28°S-28°N,
28-60°N and 60-90°N, accounting for the regional RF caused by the emissions. This regional
temperature response is calculated using a set of Regional Climate Sensitivities, $RCS_{i,l,m}$. The
$RCS_{i,l,m}$ is the unitless regional response in latitude band $m$ due to a radiative forcing in latitude
band $l$ caused by a change in species $i$, relative to global sensitivity. Hence, the $RCS_{i,l,m}$ express the
relative regional response pattern. The regional climate sensitivities are developed based on a large
set of simulations performed with the coupled atmosphere-ocean climate model GISS (Shindell &
Faluvegi, 2009; Shindell & Faluvegi, 2010).
For BC, OC, $SO_2$, $NO_x$-induced ozone and contrail-cirrus, the ARTP in latitude band $m$ at time $H$
following a pulse emission  is calculated as:
$$ARTP_{i,r,m}(H) = \sum_l \frac{RF_{i,l,r}}{E_{i,r}} \cdot RCS_{i,l,m} \cdot IRF(H) \qquad (4)$$
where $RF_{i,l,r}$ is the RF in latitude band $l$ caused by one year of emissions $E_{i,r}$ of species $i$ in region
$r$. The impulse response function IRF(H) is a temporal temperature response to an instantaneous
unit pulse of RF, which includes the global climate sensitivity. Here we have used the IRF of
Boucher and Reddy (2008), which gives an equilibrium climate sensitivity (ECS) of 1.06 K (W m$^{-2}$)$^{-1}$, equivalent to a 3.9 K equilibrium response to a doubling of $CO_2$. This is in the upper range
reported in the Fifth IPCC Assessment Report (Bindoff et al., 2013). Assuming that the regional
climate sensitivities scale linearly with the ECS, adopting a lower value reduces the magnitude of
temperature response, and its time evolution, but does not affect the latitudinal distribution.
Equation 4 can be used for short-lived species where $H$ is significantly longer than the lifetime of
the species (typically days to weeks). In the case of NOx-induced methane and subsequent ozone
changes, the longer atmospheric residence time demands an additional IRF that describes the
atmospheric decay of methane (IRF$_{long}$). We add:
$$IRF_{long}(t) = e^{-t/\tau} \tag{5}$$
where $\tau$=11.3 years is the adjustment time for methane for this model run. The ARTP$_{long}$ is then
calculated following:
$$ARTP_{i,r,m,long}(H) = \sum_l \int_o^t \frac{RF_{l,r}}{E_{i,r}} \cdot IRF_{long}(t) \cdot RCS_{i,l,m} \cdot IRF(H-t)dt \tag{6}$$
The net ARTP is the sum of contributions given by Equations 4 and 6.
The $RCS_{i,l,m}$ used in the emission metric calculations are summarized in Table SI3. For OC, sulfate,
nitrate, contrail-cirrus and methane (plus methane-induced ozone changes) we use the $RCS_{i,l,m}$ of
the mean of the responses to $CO_2$ and sulfate, as tabulated in Shindell and Faluvegi (2010). For
BC and NOx-induced ozone change the respective $RCS_{i,l,m}$ from Shindell and Faluvegi (2009) (and
tabulated in Collins et al. (2013)) are used.
The temperature response per unit RF can differ between different forcing mechanisms, i.e.,
mechanisms can have their own specific climate sensitivity parameter. This is often expressed as
an efficacy, defined as the ratio of the climate sensitivity parameter for a given forcing agent to
that for a given change in $CO_2$ (Hansen et al., 2005). The efficacy depends primarily on the spatial
distribution of the RF, both in the horizontal and vertical. The $RCS_{i,l,m}$ implicitly include
differences in efficacy of individual components arising from the horizontal forcing distribution
(to the extent that the driving processes are accounted for in the underlying climate model
simulations). The $RCS_{i,l,m}$ are established for the four forcing agents BC, ozone, sulfate and $CO_2$.
Contrail-specific regional sensitivities do so far not exist. Two studies have indicated that the
efficacy of line-shaped contrails may be as low as 0.3-0.6 (Ponater et al., 2005; Rap et al., 2010).
However, little or no information about the efficacy of contrail-cirrus and the dependence of the
climate sensitivity parameter of contrails on the horizontal forcing distribution exist. It should also
be noted that efficacies from the small sector-specific forcings can currently only be derived by
applying large scaling factors to produce forcings large enough to give a significant response in
the climate models. This adds an additional uncertainty to deriving reliable $RCS_{i,l,m}$, in particular
for contrail-cirrus due to the saturation effects. Using the average sensitivities of sulfate and $CO_2$
to calculate the ARTPs of contrail-cirrus allows us to account for a broader set of aviation-induced
forcing mechanisms in our analysis, and these $RCS_{i,l,m}$ include both a longwave absorption and a
shortwave scattering component. The efficacy of scattering aerosols and greenhouse gases is also
likely less dependent on altitude than for absorbing aerosols. However, we recognize that there
can be uncertainties associated with this approach that can presently not be quantified. As for
estimates of efficacies at the global scale, such as those given by Ponater et al. (2005) and Rap et
al. (2010), these can be included in the metric application as a scaling factor, as discussed in Sect.
3.3. However, presently few studies have investigated the efficacy of aviation-induced forcing
mechanisms.     The dependence of the climate sensitivity parameter on the altitude of the
perturbation is discussed in more detail in Sect. 3.4. We also explore potential uncertainties in our
analysis arising from such vertical dependence by comparing the temperature responses estimated
using the $RCS_{i,l,m}$ with temperature simulated by three additional climate models (Sect. 2.3).
Emission metrics are given on a per unit emission basis. However, for contrail-cirrus it is not clear
how to pose the metric since no direct correspondence between an emission and the consequent
forcing exists in this case. In order to provide consistent mass-based metrics for all components,
we adopt the same approach as Fuglestvedt et al. (2010) and calculate the contrail-cirrus GWP,
GTP and ARTP per unit $CO_2$ emitted. However, as also discussed in Fuglestvedt et al. (2010), an
alternative is to relate the contrail-cirrus forcing to the distance flown. This approach may be more
consistent with the way aircraft generate contrails and here we also provide metrics on a per km
basis. Both approaches are problematic when applying the methodology to future air traffic
scenario which likely include fuel efficiency improvements. An increase in fuel efficiency causes
a higher probability of contrail formation and at the same time a decrease in $CO_2$ emissions.
Therefore, contrail-cirrus RF per $CO_2$ emission would increase strongly, whereas contrail-cirrus
RF per flight distance would increase less so.
In the following, we present emission metrics and calculate temperature changes for time horizons
of 20 and 100 years after a one-year pulse of present-day aviation emissions. Real world emissions
are of course not pulses, but rather change over time following the development in economic
activity, technology and regulations. However, pulse based emission metrics can be used to
quantify the net difference between two emission scenarios since any scenario can be viewed as a
series of pulse emissions and analyzed through convolution (Eq. 7 below). Metrics for pulse
emissions are also useful in themselves for illustrating the temporal behavior of various species.
Realistic emissions will be continuous, leading to different relative contributions of short- and
long-lived, warming and cooling species over time. Through the use of convolution, our metric
framework can be used to estimate the temperature impact following any emission scenarios $E_{i,r}(t)$.
For instance, the regional temperature response in latitude band $m$ for species $i$ for a scenario is
the convolution of the emission scenario and the ARTP for a pulse emission (Aamaas et al., 2013):
$$\Delta T_{i,r,m}(t) = \int_0^t E_{i,r}(t') \cdot ARTP_{i,r,m}(t - t')dt' \tag{7}$$
For most short-lived species, the result will be a scaling of the ARTP value for a certain time
horizon. However, this is not the case for NOx, where the different time scales of the warming
ozone effect and cooling effects linked to methane results in a change of the sign of the emission
metric over time (as illustrated for GTP by Aamaas et al. (2016)).
To establish ranges in the global mean temperature change after 20 and 100 years due to
uncertainty in RF and ECS, we perform a Monte Carlo analysis with 100 000 draws. Each RF
mechanism is treated as a random variable, following a probability density function (PDF) defined
using estimates from the existing literature. For the aerosols, we use the multi-model results from
the AeroCom Phase II experiment (Myhre et al., 2013a), while for $CO_2$ and the NOx-induced
changes in ozone and methane, we use the uncertainties from the IPCC AR5 (Myhre et al., 2013b).
The NOx-induced ozone and methane impacts are assumed to be dependent and a PDF for the net
RF is established. Relative uncertainties are given in Table SI4. For contrail-cirrus we infer a
lognormal distribution using the best estimate of 0.05 W m$^{-2}$ and 90% confidence interval of [0.02,
0.10] W m$^{-2}$ based on IPCC AR5. The distribution of the total RF is derived by summing the PDFs
of individual mechanisms. This approach assumes that the forcing uncertainties are independent.
We also adopt a lognormal distribution for the ECS and assume a best estimate of 3 K for a
doubling of $CO_2$, and an upper and lower value of 1.5K and 4.5K (Bindoff et al., 2013). Ranges
are given at the 1 SD level (16% and 84% percentiles).

An additional source of uncertainty is the regional climate sensitivities. A full set of $RCS_{i,l,m}$ have
so far only been estimated with one climate model and it can be expected that they are likely model
dependent. When compared with the response to historical aerosol forcing in several other climate
models, the sensitivities seem fairly robust (Shindell, 2012). Two studies have also repeated the
BC experiments from Shindell and Faluvegi (2009) for BC with similar findings in terms of spatial
distribution of forcing and temperature response (Flanner, 2013; Sand et al., 2013). However, this
evaluation is limited and a formal quantification of the uncertainty or model dependence is
currently not possible.

**2.3 SIMULATED TEMPERATURE RESPONSE**
To evaluate the application of the regional climate sensitivities in the context of aviation RF, we
compare temperature responses estimated using the ARTP concept with temperature response
patterns simulated by three climate models: the NCAR Community Earth System Model
(CESM1.2) (Hurrell et al., 2013), HadSM3 (Williams et al., 2001) and ECHAM (Stenke et al.,
2008). Simulations with the two latter models were performed within the EU project QUANTIFY
(Ponater et al., 2009) and results used in Lund et al. (2012). Simulations with the CESM1.2 are
performed for this study using the aviation ozone concentration perturbation from OsloCTM3. In
order to obtain a statistically significant response to aviation ozone in the model, the perturbation
is scaled by a factor 40 (similar factors were applied in the HadSM3 and ECHAM simulations, see
Lund et al. (2012) for details). We run a four member ensemble of 60 years, using the last 30 in
the analysis. The statistical significance is assessed using the False Discovery Rate approach (FDR)
(Wilks, 2006). Here we focus on regional patterns of temperature response, but we recognize the
potential non-linearities that may arise when scaling of this magnitude is applied (e.g., Shine et al.
(2012)) and uncertainties should be kept in mind when considering the absolute magnitude of
temperature responses. Figure SI 1A shows the zonal annual mean ozone concentration change
caused by global aviation NOx emissions from the OsloCTM3 (i.e., before scaling), while Fig. SI
1B shows the annual mean CESM2.1 temperature response to the scaled ozone perturbation
(hatching indicates statistical significance at the 0.05 level).
We also compare temperature responses to aviation BC simulated by HadSM3 using the same
model configuration as given in Shine et al. (2012).
For comparison with climate model results, we use the regional climate sensitivities to estimate
the regional equilibrium temperature response ($\Delta T_{i,r,m}$) to a constant forcing  following Eq. 6 of
Shindell (2012)

$\Delta T_{i,r,m} = \sum_l RF_{l,r} \cdot RCS_{i,l,m} \cdot ECS$ \hfill (8)
and adopting the equilibrium climate sensitivity (ECS) inherent in the IRF from Boucher and
Reddy (2008) described above.

**3 RESULTS AND DISCUSSION**
**3.1 GLOBAL EMISSION METRICS**
Tables 1 and 2 summarize the 20 and 100 year GWPs and GTPs of global and regional aviation
emissions, respectively, given relative to $CO_2$ using the $CO_2$ impulse response function ($IRF_{CO2}$)
from Joos et al. (2013). The global GWPs and GTPs are not the main focus of our study, but are
included and briefly described for comparison with previous estimates. Our emission metrics do
not account for climate-carbon feedbacks. If included, such feedback could increase the relative
importance of non-$CO_2$ species (e.g., Gasser et al. (2017)).
Our GWPs for the net effect of global aviation NOx are somewhat higher than the range estimated
by Skowron et al. (2013) using several different aviation emission inventories in a single model
and Myhre et al. (2010) based on multi-model results (Table 1). They also fall in the upper end of
the range reported by Fuglestvedt et al. (2010). The NOx GTPs fall at or in the positive end of
reported ranges. A number of factors can contribute to difference in the metric values, including
differences in input radiative forcing, treatment and inclusion of methane-induced changes in
ozone and stratospheric water vapor, and differences in the parameters of the $IRF_{CO2}$. Our estimates
also include the cooling effect from NOx-induced formation of nitrate aerosols, which has to our
knowledge not been accounted for in any previous aviation GWP and GTP estimates. The
estimated contrail-cirrus GWPs and GTPs are similar to those given in Fuglestvedt et al. (2010).
However, values are not directly comparable as we consider the combined RF of contrail-cirrus
(i.e., young line shaped contrails and those cirrus originating from them, and their associated
optical depth) and include the feedback of natural clouds in the present analysis. The RF of

contrail-cirrus was shown to be 9 times higher than the RF of line shaped contrails when assuming constant optical depth (Burkhardt & Kaercher, 2011). Further differences arise from the use of different $IRF_{CO2}$. Our GWPs and GTPs for the direct effect of BC and sulfate aerosols are higher than those derived by Fuglestvedt et al. (2010) by a four (a factor two for BC GTPs). However, the values from Fuglestvedt et al. (2010) are not specific for aviation emissions, but based on input multi-model mean RF from all anthropogenic emissions (Schulz et al., 2006).

Quantifying the GTPs and GWPs of regional aviation emissions allows us to capture how equal emissions in different locations can have different impacts on the atmospheric concentrations and RF, and in turn on global climate. Our calculations reveal considerable differences between regions for all species, and both metrics and time horizons, with a factor 2-4 (and higher for nitrate) difference between the highest and lowest metric value (Table 2). For the aerosols we generally find the largest magnitude GWPs and GTPs for South Asia (SAS) emissions, followed by South America and Africa (SAF) or South Pacific Ocean (SPO). The high values for the SAS region reflects a relatively long lifetime of the aerosols here compared to other emission regions. This, in turn, is likely caused by a combination of the underlying distribution of emissions, which is dominated by emissions at high altitudes (i.e., few flights landing or departing within the region), where conditions are drier (i.e., less wet scavenging of the aerosols). For NOx, the values are highest for SPO, while for contrail-cirrus we find high values for aviation over NAM and Europe (EUR), where conditions for contrail formation is prevalent (e.g., Burkhardt et al. (2008); Irvine and Shine (2015)) and for SAF (see more detailed discussion in Sect. 3.2). From a policy perspective, knowledge of such regional differences is important if metrics are used to quantify the climate impact of emissions or emission changes in cases where there is a simultaneous change in the geographical emission distribution, or if used to evaluate the effect of implementing measures to reduce emissions in different regions.

While several studies have estimated GWPs and GTPs for global aviation NOx emissions, as discussed above, few have produced estimates for regional aviation emissions. Köhler et al. (2013) quantified the climate impact of aviation emissions in North America, Europe, India and China. The reported GWP(20) agree within 10-30 percent with estimates for all regions in the present analysis, while our GTP(20) values are about 50 percent lower in absolute magnitude for all emission regions, ande GWP(100) and GTP(100) for NAM and EUR are 50-100 percent higher

than Köhler et al. (2013) estimates. Again it should be noted that these difference can be caused
by a number of factors. Moreover, because the two studies use differently defined emission source
regions, only a rough comparison is possible.

**3.2 REGIONAL EMISSION METRICS**
While GWPs and GTPs illustrate how equal emissions in various regions can have different
impacts on global climate, they can naturally not inform us of the actual regional distribution of
impacts. The ARTP allows us to estimate temperature impacts with at least some spatial
information.
Figure 2 shows the ARTP with a time horizon of 20 years (ARTP(20)) for BC, OC, $SO_2$ and
contrail-cirrus for each emission source region, and the ARTP(20) and ARTP(100) of aviation
NOx. We do not show ARTP(100) for aerosols and contrail-cirrus here. The absolute values decay
strongly over time, but the latitudinal patterns will be identical on both time horizons. Results for
global aviation emissions are given in the SI, as are contrail-cirrus metrics on a per km-basis.

For OC and $SO_2$ (Fig. 2B and C), we calculate the highest magnitude ARTP(20) (i.e., temperature
impact per unit emission) for aviation in SAS in all latitude bands except 90-28°S, where values
for SAF and SPO are slightly higher. Excluding the SAS region, aviation in EUR and NAM give
the highest temperature impact per unit emission in the 28-60°N and 60-90°N regions, which is
also where the corresponding RF is strongest (Table SI2). This is unsurprising given that these
species are short-lived and the forcing they exert are largely localized to the emission region.
However, using the ARTP reveals important differences between the latitudinal distribution of RF
and subsequent temperature response. In given latitude bands, the temperature impact can be
considerably stronger than the RF in that band suggests, and can extend to the opposite hemisphere
to where the emissions occurred. This can be seen by comparing the latitudinal distribution of the
RF values given in Table SI2 with that of the ARTPs (note that absolute magnitudes are not directly
comparable since the ARTPs are given per unit emissions and as a function of time). Applying the
$RCS_{i,l,m}$ given in the coarse latitude bands smooths out the impacts such that there is less latitudinal
variation in the temperature responses than in the RFs. This illustrates the dependence of
temperature response on both forcing exerted locally and on remote impacts through large-scale
circulation impacts and feedbacks in the climate system.

The latter effects are also very important in the case of BC (Fig. 2A). Again, the ARTP(20) is
highest for aviation in SAS, while the difference between remaining regions is smaller than for
OC and $SO_2$ in most latitudes bands. In the 60-90°N region, aviation in the Southern Hemispheric
regions cause the highest temperature per unit emissions, despite being far removed from the
Arctic. In the GISS results, the Arctic temperature response to local (i.e., within Arctic) forcing is
in fact negative (Shindell & Faluvegi, 2009). This local cooling effect applies to BC changes in
the mid to upper Arctic troposphere, which is where aviation is most important. Aviation BC
emissions in EUR and NAM are more easily transported into the Arctic region and hence induce
a stronger local forcing and in turn a larger surface cooling. The net effect of aviation in these
regions on the Arctic is still a warming, but this warming is partly offset by the cooling contribution
from within Arctic RF. In contrast, aviation BC emissions in SAS, EAS, SAF and SPO have less
potential for long-range transport to the Arctic, but the remote BC forcing warms the Arctic
through transport of energy. In terms of mitigation, these results underline that if the goal is to
limit temperature increase e.g., in the Arctic, it is necessary to go beyond radiative forcing as an
indicator and to also consider the impact of emission in more remote regions. This feature has been
illustrated also for other sectors and regions (Collins et al., 2013; Lund et al., 2014).

For contrail-cirrus (Fig 2D), the ARTP(20) for aviation in EUR and NAM is substantially larger
in the 28-60°N and 60-90°N latitude bands than for other sources regions considered in this study,
while the difference between source regions is less pronounced in the Southern Hemisphere
latitude bands. There are two main competing processes at play. Contrail formation is generally
more prevalent in the extratropics due to lower temperatures at flight levels than in the tropics and
may persist longer due to larger probability of ice supersaturation. An upward shift in the flight
level in the tropical troposphere increases the probability of contrails formation and ice
supersaturation (Burkhardt et al., 2008). Local peaks of ice supersaturation are also found in the
tropics (Irvine & Shine, 2015), in fact the probability of ice supersaturation is highest in the upper
tropical troposphere (Lamquin et al., 2012). Furthermore, once contrails have formed, the optical
depth of contrail-cirrus is higher in the tropics due to the larger amount of water vapor available
for deposition. This higher optical depth in the tropics and the consequently higher RF has also
been found in contrail-cirrus simulations (Burkhardt & Kaercher, 2011). However, none of our
source regions cover only the tropics. In the SAS region, the air is mostly too warm for contrail
formation. However, if contrails were present here, their radiative forcing efficiency would be high.
Due to the competing short- and long-wave effects, there can be important diurnal and seasonal
variability in the net impact of contrail-cirrus (e.g., Stuber et al. (2006); Bock and Burkhardt
(2016b)). The diurnal effects depend on assumptions about the contrail-cirrus lifetime and were
shown to be small (Newinger & Burkhardt, 2012) when using the contrail-cirrus parameterization
of Burkhardt and Kaercher (2011). This effect is not captured in our analysis using annual mean
RF as input.

It should be emphasized that the contrail-cirrus metrics are suitable for the average of the present-
day aircraft fleet. Their application would not be appropriate if there are significant changes in
routes and flight altitudes. Furthermore, future changes in climate may alter the meteorological
and dynamical conditions, and hence affect the potential for contrail-cirrus formation in a given
region (Irvine & Shine, 2015). As discussed in Sect. 2, several factors contribute to uncertainty in
the emission metrics, and should be kept in mind in further applications.

The ARTP of aviation NOx (Fig. 2E, F) is separated into contributions from ozone, methane and
methane-induced ozone, as well as the direct effect of nitrate aerosols. The stars indicate the net
NOx effect. The ARTP(20) is negative in all but two cases, i.e., the net effect of one year of
aviation NOx emissions is a cooling on this time scale, dominated by the NOx-induced methane
loss. However, it is important to note that the sign and magnitude of the net NOx effect is very
sensitive to the choice of time horizon due to the very different time scales on which the ozone
and methane contributions act. In particular, during the first decade after emission, the strong but
short-lived warming from ozone dominates, resulting in a net positive effect (see also Fig. 6B of
Fuglestvedt et al. (2010)). Moreover, the NOx ARTP is also influenced by the spatial patterns of
ozone and methane. Due to the shorter lifetime, the aviation-induced ozone perturbation is more
heterogeneous than the methane concentration change, and more confined to the emission source
region. The choice of time horizon may therefore affect both the net NOx effect and the relative
importance of source regions. Both the impact of changes in ozone and methane from a pulse
emission on NOx decays strongly over time, as reflected by the much smaller ARTP(100). While
the methane cooling remains important on longer time scales, the absolute magnitude diminishes
strongly towards a time horizon of 100 years. For all source regions, the competing effects of
ozone warming and methane cooling over time results in a small, but net positive global mean
effect of aviation NOx on a 100 year time horizon. In the 90-28°S latitude band the ARTP(100) is
positive for emissions occurring in the Southern Hemisphere, but negative for emissions in the
Northern Hemisphere, as the latter causes a much smaller ozone concentration change in this
latitude band and the methane cooling becomes relatively more important over time. A similar
response is seen in the 60-90°N latitude band.

**3.3 REGIONAL CLIMATE IMPACTS OF THE PRESENT-DAY AVIATION SECTOR**
To place our climate metrics in context and illustrate further application, we apply the ARTP and
estimate the regional temperature responses over time to present-day aviation in the six emission
source regions. Figure 3 shows the temperature change (net and contribution from each species)
in each latitude band 20 and 100 years after a *one-year pulse* of present-day aviation emissions in
each source region. The current aviation climate mitigation policy is largely focused on $CO_2$. The
contributions from aviation $CO_2$ emissions are therefore added to place the impact of short-lived
and long-lived species in context. The error bars show the 1 SD ranges due to uncertainties in RF
and ECS (see Sect. 2.2). Because the same relative uncertainty is assumed for emissions in all
source regions and our analysis does not account for uncertainty in the regional climate
sensitivities, we only include the ranges in the global mean temperature change.
The majority of flights today take place over the northern mid-latitudes. As a result, the net
warming is largest in all latitude bands for emissions over NAM and EUR. On a 20 year time
horizon, the largest warming contribution from these source regions comes from contrail-cirrus
formation and, despite the highly localized RF, the temperature impact is not only limited to the
latitude band closest to where the emissions occur. The net warming is slightly offset by a small
cooling due to NOx-induced methane loss, especially in the 90-28°S and 60-90°N regions. As
pointed out above, the sign and magnitude of the net NOx effect depends strongly on the chosen
time horizon. For instance, on a 10 year time horizon (not shown here), the net NOx response is a
warming in the 28°S-28°N and 28-60°N regions for emissions in NAM, EUR and EAS. Aviation

emissions of BC are small and therefore contribute little to the net impact, despite the strong efficiency (i.e., temperature change per kg emitted).

Even on short time horizons (e.g., 20 years), the warming contribution from $CO_2$ is important. For emissions in EAS, SAF, SAS and SPO, $CO_2$ is of comparable magnitude to contrail-cirrus after 20 years. On longer time horizons (e.g., 100 years) the $CO_2$ contribution becomes dominant in all latitude bands. This has previously been illustrated for the global mean temperature impact of the sector (Berntsen & Fuglestvedt, 2008). Because the perturbation in $CO_2$ is longer-lived and well-mixed, the warming in the Southern Hemisphere becomes relatively more important compared to the other latitude bands on longer time scales. Nevertheless, for emissions in NAM and EUR, for the northern hemisphere response regions and the global mean, the contributions from contrail-cirrus remains substantial and approximately 10-20 percent of the $CO_2$ response, even on these long time scales. Figure 3 also shows that the relative importance of the source regions across latitude bands following a pulse emission changes very little over time. However, these calculations do not account for potential future changes in the geographical distribution of emissions.

The considerable uncertainty in the aviation-induced forcing mechanisms and climate sensitivity is reflected in the error bars. After 100 years, the uncertainty in climate sensitivity dominates as the relative contribution from the more uncertain, but short-lived mechanisms decays and $CO_2$ becomes more important. Note that the same relative uncertainties apply to all source regions. For contrail-cirrus, an additional source of uncertainty is the efficacy. As noted in Sect. 2.2, studies indicate that the efficacy of contrail-cirrus may be lower than one. Because only two estimates exist in the literature, efficacy is not included in present analysis. However, adopting a spatially uniform efficacy of e.g., 0.6 (Ponater et al., 2005) would result in a 40 percent lower contrail-cirrus impact across all latitude bands.

Our study focuses on the pulse based emission metrics and does not consider the future temperature impact of aviation following *emission scenarios*, which would change the timescale of the response and the relative importance of short- and long-lived species over time. As described in Sect. 2.2, our pulse based emission metrics can be used in further studies to investigate the regional temperature impacts following more realistic emission scenarios. For instance, as the simplest form of scenario, one could assume that emissions are kept constant at the present-day level. In

this case, the contributions from short-lived species such as contrail-cirrus, ozone and sulfate
would quickly become sustained at a constant level rather than decay towards zero and the
warming from $CO_2$ would gradually accumulate (e.g., Berntsen and Fuglestvedt (2008)). The
impact of contrail-cirrus may even increase if emissions are kept constant while fuel efficiency is
improved. The temporal behavior of total net temperature response, as well as the net NOx effect,
would differ notably from the pulse emission case.

**3.4 EVALUATION**
Several studies have calculated ARTPs for emissions from specific sectors or regions (Collins et
al., 2013; Lund et al., 2014; Sand et al., 2016; Stohl et al., 2015). Stohl et al. (2015) also compared
the estimated regional temperature responses to short-lived climate pollutants with those simulated
by several climate models. However, these studies focus only on surface sources and the evaluation
may not be valid for aviation. The regional climate sensitivities that form the basis for the ARTP
calculations are derived from simulations with only one climate model. Moreover, the sensitivities
are representative of the response to a vertical forcing profile resulting from total anthropogenic
emissions, i.e., one that in many regions differ considerably from those induced by the mainly
high-altitude aviation emissions. Several recent studies have found a strong vertical sensitivity in
the BC forcing-response relationship, with decreasing efficacy with altitude (Ban-Weiss et al.,
2011; Flanner, 2013; Samset & Myhre, 2015). Climate model studies also indicate that the forcing-
response relationship for ozone will be dependent on both the vertical and horizontal distribution
of the ozone change (Berntsen et al., 1997; Hansen et al., 1997; Joshi et al., 2003), which in turn
also depends on altitude (e.g., Olsen et al. (2013)). As discussed in Sect. 2.2, a formal
quantification of uncertainties in the regional climate sensitivities is currently not possible.
However, in light of the potential uncertainties arising from the vertical dependence, we perform
a first evaluation of the ARTP concept in the context of aviation ozone and BC. Further evaluation,
especially of contrail-cirrus, would be valuable, but require resources beyond those available for
the currrent study.

Figure 4A shows the normalized regional temperature response to aviation ozone, as simulated by
the CESM1.2, HadSM3 and ECHAM (see Sect. 2.3) and estimated using the regional climate
sensitivities that form the basis for the ARTP concept. There are several factors potentially
contributing to differences in the absolute magnitude of temperature responses in the simulations,
including differences in the ozone concentration perturbation resulting from differences in
emissions or ozone change per unit emission, radiative efficiency and ECS. HadSM3 and ECHAM
used the multi-model average ozone concentration change resulting from year 2000 aviation NOx
emission (0.67 TgN) (Hoor et al., 2009), while this study (using CESM2.1) uses the ozone change
simulated by one model (OsloCTM3) and year 2006 aviation emissions (0.81 TgN). Based on
visual inspection, these two aviation-induced ozone concentration perturbations are quite similar,
with slightly larger perturbation at high northern latitudes in the present study. Nonetheless, here
we focus on the spatial pattern across latitude bands rather than absolute magnitudes and therefore
normalize the temperature response in each band by the respective global mean response.
The climate models and RTP agree reasonably well in the 28°S-28°N and 28-60°N latitude bands.
However, in both the 90-28°S and 60-90°N regions, the temperature response simulated directly
by the climate models is considerably higher than that estimated using the ARTP. The low ARTP-
derived temperature response in the 60-90°N region reflects the low sensitivity in the GISS model
in this latitude band to ozone forcing exerted both locally, as well as in the 28-60°N region where
the aviation-induced forcing is highest (Fig. 1 of Shindell and Faluvegi (2009)). A low, and even
negative, sensitivity to Northern Hemisphere forcing also characterizes the 90-28°S band.
The reason for the low sensitivity in the GISS simulations, or whether this is a feature specific to
this model, is not clear. It is possible that the differences between modeled and estimated
temperature response to aviation ozone forcing can be at least partly explained by vertical
variations in the response to ozone perturbations. Early work by Hansen et al. (1997) suggested a
surface cooling in response to a global near surface ozone perturbation and a maximum efficacy
around 700-800 hPa, followed by a decreasing efficacy for ozone perturbations in the upper
troposphere. The latter was supported by Joshi et al. (2003). However, the increased sensitivity to
lower stratosphere perturbations found by Joshi et al. (2003) was not seen in the Hansen et al.
(1997) results. Such uncertainties in the efficacy around the UTLS region are important in the case
of aviation.
Figure 4B compares aviation BC temperature response to that obtained from the HadSM3. The
regional distribution across latitude bands is similar for the estimated and simulated temperature
response. However, here we only have temperature response simulated by one climate model.
Given the substantial uncertainty and inter-model differences in model estimates of BC climate
impacts (Baker et al., 2015; Samset et al., 2014; Stohl et al., 2015), additional model simulations
are needed for further comparison and evaluation.
The notable decrease in BC efficacy with altitude globally, and particularly at high latitudes (Ban-
Weiss et al., 2011; Flanner, 2013; Samset & Myhre, 2015), raises the question whether using the
ARTP to estimate temperature responses to the high-altitude aviation BC forcing could result in
an overestimation of the absolute magnitude. Flanner (2013) provided vertically resolved climate
sensitivities for the Arctic temperature response to local Arctic BC forcing. Using these, Lund et
al. (2014) found important differences in the temperature response to BC from on-road
transportation in the 60-90°N latitude band compared to using the single regional climate
sensitivity derived from Shindell and Faluvegi (2009) (and used in the present analysis). At the
altitudes in the 60-90°N region where the aviation-induced RF peaks, the regional climate
sensitivity from Flanner (2013) and Shindell and Faluvegi (2009), and hence the estimated
temperature response, agree quite well. This agreement may not hold for all regions, but similar
vertically resolved climate sensitivities for other latitude bands or species do not currently exist.
Based on our analysis, some care is needed when using the ARTP in the context of aviation
emissions. Specifically, our findings suggest that the temperature response in the 90-28°S and 60-
90°N regions to aviation ozone could be underestimated by the regional climate sensitivities
currently used in the ARTP calculations. Furthermore, a possible overestimation of temperature
response to aviation BC can not be ruled out. Further work to quantify the importance of vertical
variations in forcing-response relationships and develop regional climate sensitivities based on
vertically-resolved forcing perturbations would be valuable for future use of the ARTP.

**4 CONCLUSIONS**
We have examined the impacts of aviation emissions on global and regional temperature,
characterizing them using emission metrics. We address the impacts of $NO_x$ on ozone and methane,
aerosols and contrail-cirrus formation, and consider six emission regions spanning both
hemispheres. In addition to updated emission metrics for global aviation, we present GWPs and
GTPs on 20 and 100 year time horizons for a larger set of species and regions than previous studies.
We also calculate the Absolute Regional Temperature change Potential (ARTP) for aviation,
allowing us to not only capture how equal emissions in different regions impact global climate,
but also quantify the temperature impacts on a sub-global scale.
The metric values depend significantly on emission regions. In the case of aviation aerosols, we
calculate the highest GWPs and GTPs for emissions in South Asia, followed by South America
and Africa, and the South Pacific Ocean. The strong efficiency of emissions over our South Asian
region reflects a relatively long lifetime of the aerosols here compared to other region. Our results
do not include aerosol-cloud interactions, an important limitation as recent studies suggest aviation
can potentially have strong impact through modification of both cirrus and low-level clouds;
however this contribution remains particularly uncertain. The net temperature impact over time
following a pulse emission of aviation NOx is determined by the relative importance of the cooling
and warming methane and ozone contributions, and is very sensitive to the choice of time horizon.
The net NOx ARTP is negative after 20 years and switches to a small net warming on a 100 year
time horizon on global mean and in the latitude band closest to the where the emission occur.
Metrics for contrail-cirrus are calculated on a per unit emission of aviation $CO_2$ basis. The GWPs
and GTPs are highest for North America and Europe, where contrail-cirrus formation is prevalent.
However, once formed, contrail-cirrus in the tropics have much higher optical depth due to the
larger amount of water vapor. The metric values do not account for a lower efficacy of contrail-
cirrus that has been suggested by previous studies, but remains highly uncertain. Moreover, the
contrail-cirrus metrics would not be appropriate if there are significant changes in routes and flight
altitudes, or if changes in climate or propulsion efficiency affect the potential for contrail-cirrus
formation.

The ARTPs illustrate how the latitudinal temperature pattern can differ significantly from the
global mean, as well as from the latitudinal pattern of RF. Due to the short lifetime of many of the
aviation forcing mechanisms, the RF is typically largely confined to the latitude band closest to
where the emissions occur. However, in a given latitude band, the temperature response can be
considerably stronger than suggested by the corresponding forcing, emphasizing the importance
of both forcing exerted locally and remote impacts through large-scale circulation impacts and
feedbacks in the climate system.

While the strongest temperature change per unit aerosol and NOx emitted is found for aviation
over the South Asian region in our study, the majority of flights today take place over the northern
mid-latitudes. The net warming impact 20 and 100 years following a one-year pulse emission from
the present-day aviation sector is therefore largest in all latitude bands for emissions in North
America and Europe, with the largest warming contribution after 20 years from contrail-cirrus.
Furthermore, contributions from contrail-cirrus remain important at the 10-20 percent of $CO_2$ level
in several regions even on a time horizon of 100 years. The discussion around $CO_2$ often focuses
on its long-term impacts. Our results illustrate that while $CO_2$ is dominant on longer time scales,
it also gives a considerable warming contribution already after 20 years. Our metric framework
can also be applied to estimate future regional temperature impact of more realistic emissions
scenarios for the sector, which would influence the temporal characteristic of the response and the
relative contributions of short and long-lived species over time.
While the ARTP concept is an important and useful tool for providing first order estimates of
regional temperature of various emissions, our analysis indicate that some care is needed when it
is used in the context of aviation emissions, or more generally, in situations that differ significantly
from those used to derive the regional climate sensitivities for the ARTP calculations in the first
place. In particular, further work to quantify and account for the relationship between vertically-
resolved radiative forcing perturbations and surface temperature response is needed to allow for a
more general applicability of the concept.

## Acknowledgements

This is work funded by the US Federal Aviation Administration (FAA)/Volpe Center under the
contract no. DTRT57-12-P-80123. We thank Dr. Øivind Hodnebrog (CICERO) for contributions.

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

**Tables**
*Table 1: GWP and GTP of global aviation emissions for time horizons 20 and 100 years. Values*
*are given on a per unit aviation emission basis (per kg BC, OC and $SO_2$, respectively, while*
*contrail-cirrus is calculated per kg aviation $CO_2$, and NOx per kg N). The GTPs are calculated*
*using the impulse response function by Boucher and Reddy (2008). For comparison, aviation NOx*
*GWP and GTP values from three previous studies are also included.*

| | GWP | | GTP | |
|---|---|---|---|---|
| **Component** | **H=20** | **H=100** | **H=20** | **H=100** |
| **Contrail-cirrus** | 3.1 | 0.84 | 0.93 | 0.12 |
| **BC** | 3911 | 1064 | 1135 | 147 |
| **$SO_2$** | -559 | -152 | -162 | -21 |
| **OC** | -282 | -77 | -82 | -11 |
| **NOx** | 411 | 77 | -138 | 9 |
| **NOx** Fuglesvedt et al. (2010) | 120 to 470 | -2.1 to 71 | -590 to -200 | -9.5 to 7.6 |
| Myhre et al. (2011) | 92 to 338 | -21 to 67 | -396 to -121 | -5.8 to 7.9 |
| Skowron et al. (2013) | 142 to 332 | 4 to 60 | | |











*Table 2: GWP and GTP of regional aviation emissions for time horizons 20 and 100 years. Values*
*are given on a per unit aviation emission basis (per kg BC, OC and SO$_2$, respectively, while*
*contrail-cirrus is per kg CO$_2$, and NOx per kg N). The GTPs are calculated using the impulse*
*response function by Boucher and Reddy (2008).*

| Component | Source region | GWP | | GTP | |
|---|---|---|---|---|---|
| | | H=20 | H=100 | H=20 | H=100 |
| Contrail-cirrus | SAF | 3.6 | 0.99 | 1.09 | 0.14 |
| | NAM | 3.3 | 0.90 | 1.00 | 0.13 |
| | EAS | 1.7 | 0.45 | 0.50 | 0.06 |
| | EUR | 2.5 | 0.67 | 0.75 | 0.10 |
| | SPO | 2.3 | 0.63 | 0.70 | 0.09 |
| | SAS | 2.6 | 0.70 | 0.78 | 0.10 |
| OC | SAF | -481 | -131 | -140 | -18 |
| | NAM | -289 | -79 | -84 | -11 |
| | EAS | -283 | -77 | -82 | -11 |
| | EUR | -197 | -54 | -57 | -7.4 |
| | SPO | -419 | -114 | -122 | -16 |
| | SAS | -611 | -166 | -177 | -23 |
| BC | SAF | 5420 | 1470 | 1570 | 203 |
| | NAM | 3560 | 969 | 1030 | 133 |
| | EAS | 4170 | 1140 | 1210 | 156 |
| | EUR | 2300 | 816 | 871 | 112 |
| | SPO | 4940 | 1340 | 1430 | 185 |
| | SAS | 8250 | 2250 | 2390 | 309 |
| SO2 | SAF | -833 | -227 | -242 | -31 |
| | NAM | -550 | -150 | -159 | -21 |
| | EAS | -602 | -164 | -175 | -23 |
| | EUR | -378 | -103 | -110 | -14 |
| | SPO | -746 | -203 | -216 | -28 |
| | SAS | -1120 | -304 | -324 | -42 |
| NOx | SAF | 484 | 70 | -316 | 6.26 |
| | NAM | 280 | 48 | -126 | 5.0 |
| | EAS | 513 | 108 | -79 | 13 |
| | EUR | 210 | 37 | -87 | 4.0 |
| | SPO | 806 | 159 | -205 | 19 |
| | SAS | 695 | 137 | -176 | 16 |




**Figures**

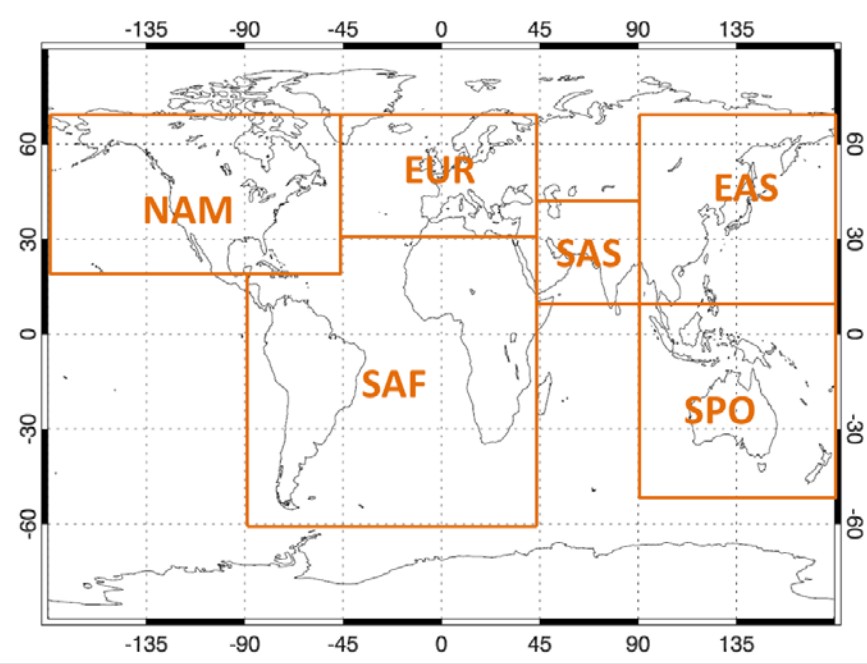

NAM = North America, EUR = Europe, SAS = South Asia and Middle East,
EAS = East Asia, SAF = South America and Africa, SPO = South Pacific Ocean


*Figure 1: Definition of emission source regions in this study.*











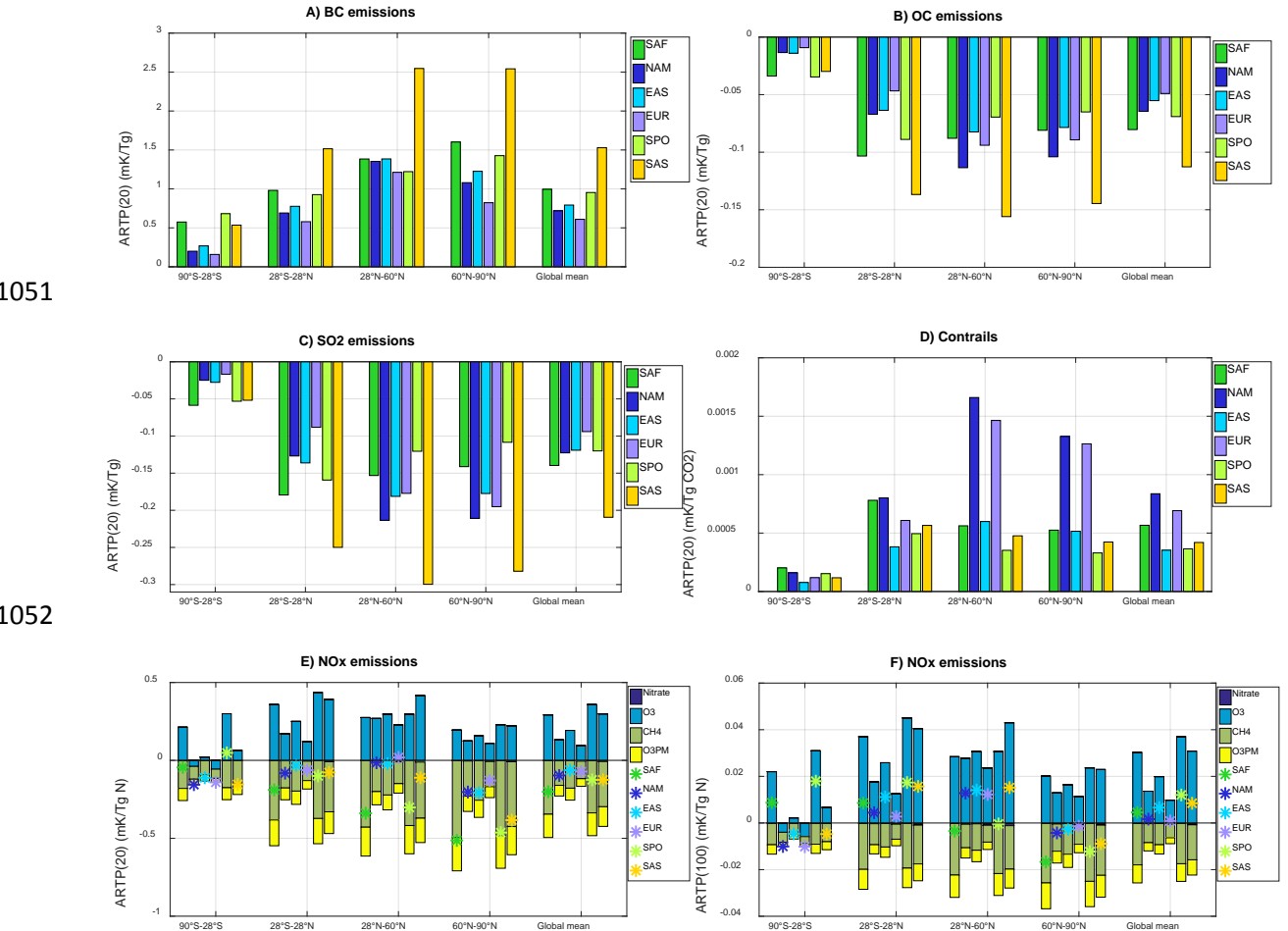



*Figure 2: ARTP(20) of aviation (A) BC, (B) OC, (C) SO₂ and (D) aviation-induced contrail-cirrus, and (E,F) ARTP(20) and ARTP(100) of aviation NOx. The asterisk in panels E and F show the net NOx effect of emissions in each source region, while the colored bars give the contributions from ozone production (O₃), NOx-induced methane loss (CH₄) (including subsequent stratospheric water vapor loss), methane-induced ozone changes (O₃PM) and NOx-induced nitrate formation.*














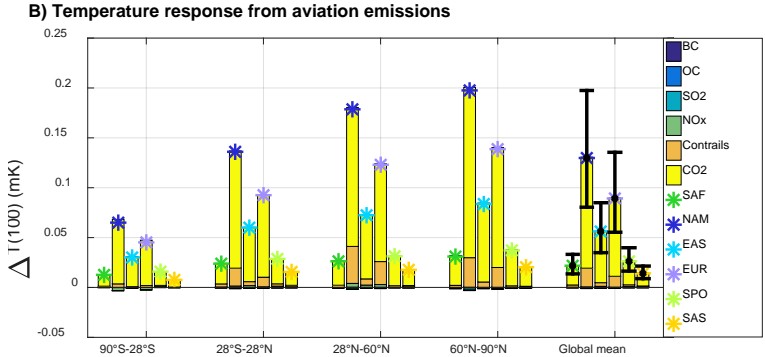



*Figure 3: Regional and global mean temperature change by species and source region after A) 20 years and B) 100 years following a one-year pulse of emission from the present-day aviation sector in each source region. The asterisk shows the net temperature response in the respective latitude band for each emission source region, while the bars show the contribution from each species to the net. Error bars show the 1 SD ranges due to uncertainties in RF and ECS.*


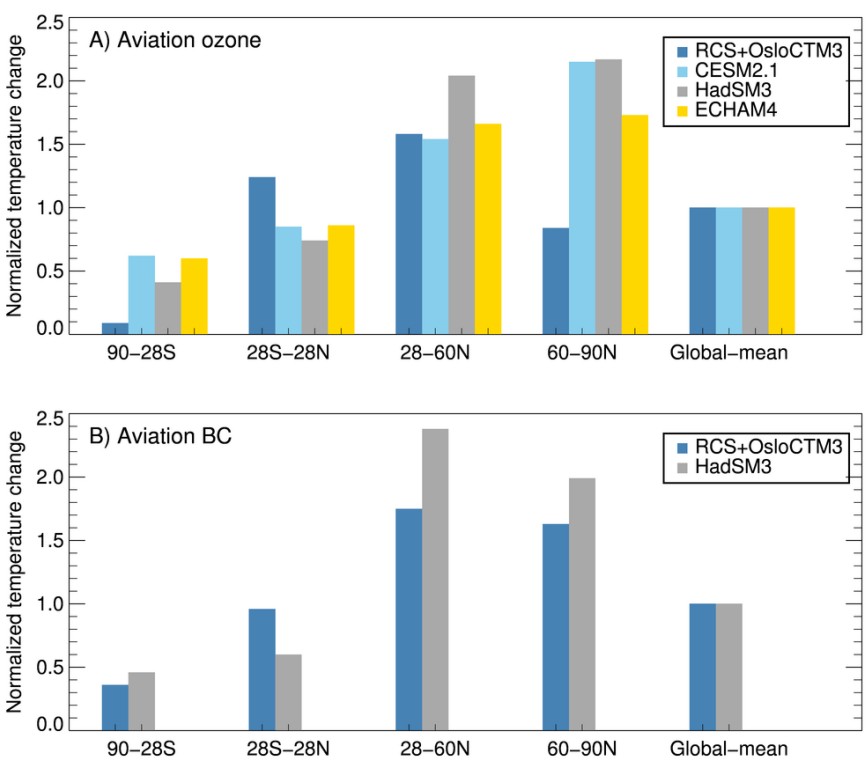


*Figure 4: A) Comparison of the regional pattern of surface temperature response to a global*
*aviation ozone perturbation as calculated using the regional climate sensitivities (RCS) from GISS*
*with RF derived from OsloCTM3 (i.e., using the ARTP concept) and simulated by HadSM3,*
*ECHAM4 and CESM1.2. Surface temperature response in each latitude band is normalized by the*
*global mean value. B) Same as A), but for BC.*