# Peer review of "Emission metrics for quantifying regional climate impacts of aviation Marianne T. Lund\*, Borgar Aamaas1, Terje Berntsen1,2, Lisa Bock3, Ulrike Burkhardt3, Jan S. Fuglestvedt1, Keith P. Shine4 1 CICERO, Center for International Climat"

_Earth System Dynamics, 2017_

## Referee Comment (RC1) · Anonymous Referee #1 · 11 Feb 2017

Review of "Emission metrics for quantifying regional climate impacts of aviation"

by Marianne T. Lund et al.

This manuscript attempts to calculate Global Warming Potentials (GWP), Global Temperature Potentials (GTP) and Absolute Regional Temperature change Potential (ARTP) for aviation emissions. This is a difficult paper to put together since there is essentially I think an important negative conclusion that this method does not work that well for aviation emissions. I would like to see a bit more uncertainty analysis to quantify where the problems are. I think a bit more discussion of the physics of the climate system would help the reader understand the problems with this method. This paper could be publishable with some important revisions.

Generally, it looks like the method described is highly model dependent, which is not

really obvious until the end of the paper, and there is not really quantification of the uncertainty. I think this could be better woven throughout. It would be nice to comment a bit more on how Stohl et al 2015 and Shindell 2012 apply to these results: I am pretty convinced that for small perturbations and for non-gaseous species like contrails that these results may not apply.

I think that this work could also use a better focus on some of the physical aspects of the climate system and the feedbacks that might or might not operate. And at a minimum I would like to see quantification of uncertainty in the different parts of the ARTP terms.

Some of the figures (e.g. Fig 3 and S1) could use some clarification as noted below.

Specific comments are below.

Page 4, L108: how does any of this work account for effects on the coupled climate system? Regional forcing can show up very differently in surface temperature depending on whether the energy goes into sensible heat, latent heat, or evaporation, and will depend on surface type and coupled modes of the climate system. For example: cooling the N. Atlantic deep water formation region with contrails may simply cool deep water and not be felt at the surface. Or it may alter the formation of deep water itself. Don't you need a coupled climate model to do this?

Page 4, L110: I'm not sure you can do this with a CTM. The circulation doesn't adjust.

Page 5, L114: again: surface temperature response depends on the surface type. Is that accounted for?

Page 5, L117: how uncertain are the regional climate sensitivities?

Page 7, L173: I do not think you can throw out this term: it is a compensating effect or feedback. You could include the reduction in cirrus as a separate forcing, but now you are overstating the contrail impact.

Page 7, L184: What are these uncertainties?

Page 7, L197: Does previous work show that this represents the physical response? I have a hard time believing it works for non-uniform emissions in the vertical, or for aviation emissions.

Page 7, L200: Are these done for contrails? This is a cloud forcing, which is very different than other forcings. For example: contrail forcing is a SW and LW forcing that depends strongly on surface albedo. Do these sensitivities take account of that? If not, I'm not sure the method is valid.

Page 8, L202: What does the statement in parentheses mean? What is a component? If the time is longer than the lifetime for a pulse emission, isn't the forcing gone?

Page 8, L207: This presumes that all RF is created equal with respect to sensitivity. I think this is valid for GHG in the IR. I'm not sure if it is valid for SW effects, and probably not for clouds.

Page 9, L229: So you are trying to put these uncertainties into 'efficacy', which would effect the contrails by a factor of 2, but then ignore it? I don't think that is appropriate. Efficacy might be a way to address some of the issues I am referring to with contrails in this context.

Page 9, L241: There are emissions scenarios for aviation that specifically address the non uniform growth of emissions and the efficiency improvements. Why not use them? There are already RF estimates in the literature with these scenarios.

Page 10, L270: I do not think the climate system is linear enough to small perturbations to make this a valid assumption. Can you provide a reference demonstrating that multiplying a forcing by an order of magnitude or more yields a linear scaling to the response to the small perturbation in a coupled climate model? I see now there is a reference below. Is it valid?

Page 10, L274: But isn't the temperature response the heart of this paper? Yet it is to

be used 'with care'? I think you might need to quantify the potential uncertainties here.

Page 10, L278: I do not think figure S1 is correct. I do not see how a 2ppb ozone perturbation causes a 1K surface temperature change in Figure S1. Please clarify or justify that.

Page 11, L287: But I thought above the forcing was an impulse, I.e. Non- constant?

Page 11, L294: This is confusing regarding emission units: since they are rationed to CO2, shouldn't they be relative to aviation CO2, then the comparison with aviation CO2 is explicit. Pardon me if that is not the appropriate definition, but I'm not sure how to interpret the very large BC GWP number.

Page 11, L305: But does this treat compensating LW and SW effects which are a function of surface type?

Page 12, L323: Dry conditions in S. Asia? Really? In some seasons. What about the summer monsoon?

Page 13, L345: I'm not sure I agree with this. How do you deal with non local energy balances in each region?

Page 13, L364: Is it valid to apply RCS from one model to CTM results from another model? They might be vastly inconsistent, particularly in remote regions.

Page 13, L370: I'm not sure the remote effects being larger make physical sense. See next comment.

Page 14, L378: Here it looks like the GISS results are convolving the effect of transport with the effect of aerosols: they co-vary, but the aerosols do not cause warming. They just come in with warmer air. I think this highlights some of the problems with this methodology. I think you draw the wrong conclusion here.

Page 14, L390: Unless emissions occur in the stratosphere, in which case contrail formation drops.

Page 15, L406: what about significant changes in meteorology: warming reduces contrails, and also changes the tropopause height.

Page 16, L454: I'm not sure exactly what is being plotted in figure 3: please explain further and in a more detailed caption. Temperature change in each region by different emissions in each region? Are the asterix just a label for the bars or does their vertical position mean anything.

Page 17, L484: If previous studies only did surface emissions (e.g. Stohl et al 2015 and Shindell 2012), then how will some of the terms from these models be valid (e.g. GISS study with RCS I think it is)?

Page 18, L511: I could also argue using this figure that the ARTP concept only works within the main emissions region, and is off by 50% or more relative to a physical calculation.

Page 19, L528: I think this whole analysis highlights that the ARTP concept is very model dependent and not a general concept because of strong dependencies you have identified here.

Page 19, L546: I think you are going to have the same problem with contrails, which also have strong vertical effects, and regional forcing dependent on the climate system itself.

Page 19, L553: Can you do an uncertainty analysis here? Where are the largest uncertainties and model dependencies in the analysis.

Page 21, L589: Where do you demonstrate that the response is stronger than the forcing? And how does this analysis include effects of the large scale circulation and feedbacks? You are breaking most of those feedbacks with use of a CTM and unitless response functions from another model.

Page 21, L606: Add to this some scaling by a factor of 40 for emissions in here somewhere, and I think this is problematic.

Page 21, L608: At least I agree with the negative conclusion here: it is not clear to me that this concept has a lot of utility for aviation. Especially since you have left out aerosol indirect effects. Anything that involves clouds I think is highly problematic using this method. Would like to see a bit more uncertainty analysis to quantify where the uncertainty lies.

Page 28, L882: What are the units of BC? Per unit of fuel or something else? Not clear from the caption.

Page 32, L922: What are the units here? Mili-kelvin? Over what area? Do the regional colors (asterisk) correspond to the emissions colors? I'm not sure what is being plotted here.

---

## Referee Comment (RC2) · Anonymous Referee #2 · 16 Mar 2017

This manuscript provides calculations of GWP and GTP for aviation emissions, focusing on aerosol direct radiative forcing impacts, GHG impacts, and impacts via contrail-cirrus formation. The authors also project temperature estimates using ARTPs based on published regional climate sensitivities, and evaluate these in comparison to their own estimates computed using 1-3 different models. The work considers emissions from 6 aggregate continental-scale regions, and climate impacts across 4 latitudinal bands.

Overall, the results of this type of study are potentially valuable for estimating the climate impacts of current and future aviation scenarios. The evaluation of the ARTP concept itself, for a vertically distributed source, is also of interest on its own. The paper is well organized and generally easy to follow. The manuscript will be suitable for publication after addressing a few comments below, most of which are made in order to provide some clarifications or additional information that will make the results presented here more broadly understandable and applicable.

Major Comments:

142: The RF kernels of Samset and Myhre (2011) are valuable because they are vertically distributed. But they are also limited in their spatial coverage. How do the authors map between the regions in their study and those in Samset and Myhre (2011)? For example, it would seem the latter does not provide any results for the author's SAS and SPO regions. A more complete 2D spatial mapping of aerosol direct radiative forcing efficiencies is provided in Henze et al. (ES&T, dx.doi.org/10.1021/es301993s, 2012). Perhaps results from these two studies could be combined to provide a more complete analysis? Or at least findings from the latter could be used to provide some sense of the uncertainty involved in using only the Samset and Myhre regions as the basis for the present work.

180-185: This argument feels a bit thin, given that some aspects of aerosol cloud interactions are at least better known that others. At the very least, could uncertainties owing to these processes be carried through the calculation, so that we know when uncertainties in these effect may alter the sign of the next outcome?

For Fig 4: why compare ARTP(20) and ARTP(100), when a more direct and fundamental comparison would be to just consider the RCS's? The RCS is what other people will need, if they are to use the results from this study themselves to calculate ARTPs. At the very least it would be quite useful to compare the RCS values in addition to the existing figures using ARTP in particular years.

General: For other people to make use of these results, it is useful to provide more information on the aviation emissions used in this study. The authors should provide a table of emissions by species and region, and they should provide separate total for emission by takeoff vs cruise altitudes. While it would be great if they could provide metrics broken down by the later category as well, I'd guess that would involve repeating a lot of calculations. But at least providing the details of the inventory they used would allow future users to be able to scale evaluations of the climate impacts of their own inventories accordingly, given some knowledge of how the authors' inventory was distributed vertically.

Minor comments:

35-38: True, but this is evident from the fact that the RCS's in e.g. Shindell 2012 are not uniform. So it is a bit odd to place this in the abstract, although I agree the application does being attention to the issue.

40-41: The feels a bit obvious (biggest emissions have the biggest impact) — would discussing the impact per emission be of more interest?

66: This statement is missing references.

78: The phrase "in a grid cell" is vague (we don't know yet how big your model grid cells are) and also ambiguous with regards to whether you are referring to grid-scale changes in temperature or grid-scale changes in emissions.

83-84: Can the authors reference any in particular?

91-93: How much uncertainty / error can this aggregation lead to?

105-110: See also Sand et al., Nature Climate, doi:10.1038/nclimate2880, 2015.

183: I don't understand what mechanisms this refers to. Please be more specific and provide references.

Table 1: could you list NOx in the first half at the bottom the list of species, so that it is easier to compare these numbers to the results for NOx from other studies listed below?

Table 1: I must be missing something — the GTP and GWP metrics are computed by emitted species (i.e., SO2 instead of sulfate), yet the authors report values for nitrate

(Table 1), and these are reported separately from NOx emissions, even though nitrate is formed secondarily in the atmospheric from NOx. Can the authors please explain this more?

Fig 2: Please explain the difference between the color bars vs star points in panels E and F, and define O3 vs O3PM in the figure caption itself.

381: Also Lacey et al. (PNAS, doi:10.1073/pnas.1612430114, 2017) used ARTPs to investigate this for cookstove emissions.

132: Is this perturbation positive or negative? Does it make a difference, for SO2 and NOx?

Fig 2: Given the factor of 0.5 in Eq 2, why isn't O3PM 50% that of the CH4 in panels E and F? Is this a consequence of the spatial re-scaling from Fry 2012? If so, I would have expected it to be less than 50% in some regions and greater than 50% in others.

222: Why not use the RCS for sulfate for sulfate, rather than for the mean of CO2 and sulfate? What RCS is used for nitrate (although not clear how nitrate is treated anyways)?

492-497: This statement is a convolution of two issues that could be separated, which are that the RF of O3 per ppb is horizontally and vertically variable, and that the climate response to this RF is also variable.

536-538 and 550-551: Fig 4 only shows the normalized results, so it is hard to know how much of an overestimate the authors are talking about here. Can they also provide the absolute results?

Did the authors consider using ARTPs for the land-only response from Shindell 2012?

Technical corrections:

87: as a bridge

133: sulfer dioxide

137: each region are

303: in the present analysis we

---

## Author Comment (AC1) · 30 Apr 2017

**Response to review of "*Emission metrics for quantifying regional climate impacts of aviation*" by Marianne T. Lund, Borgar Aamaas, Terje Berntsen, Lisa Bock, Ulrike Burkhardt, Jan S. Fuglestvedt and Keith P. Shine**

We thank the reviewer for the careful review and useful comments and suggestions. Responses to individual comments are given below.

**Anonymous Referee #1**

This manuscript attempts to calculate Global Warming Potentials (GWP), Global Temperature Potentials (GTP) and Absolute Regional Temperature change Potential (ARTP) for aviation emissions. This is a difficult paper to put together since there is essentially I think an important negative conclusion that this method does not work that well for aviation emissions. I would like to see a bit more uncertainty analysis to quantify where the problems are. I think a bit more discussion of the physics of the climate system would help the reader understand the problems with this method. This paper could be publishable with some important revisions.

Generally, it looks like the method described is highly model dependent, which is not really obvious until the end of the paper, and there is not really quantification of the uncertainty. I think this could be better woven throughout. It would be nice to comment a bit more on how Stohl et al 2015 and Shindell 2012 apply to these results: I am pretty convinced that for small perturbations and for non-gaseous species like contrails that these results may not apply.

I think that this work could also use a better focus on some of the physical aspects of the climate system and the feedbacks that might or might not operate. And at a minimum I would like to see quantification of uncertainty in the different parts of the ARTP terms.

Some of the figures (e.g. Fig 3 and S1) could use some clarification as noted below.

The general points are addressed through responses to the following specific comments, where the issues are raised again.

Specific comments are below.
Page 4, L108: how does any of this work account for effects on the coupled climate system? Regional forcing can show up very differently in surface temperature depending on whether the energy goes into sensible heat, latent heat, or evaporation, and will depend on surface type and coupled modes of the climate system. For example: cooling the N. Atlantic deep water formation region with contrails may simply cool deep water and not be felt at the surface. Or it may alter the formation of deep water itself. Don't you need a coupled climate model to do this?
The ARTPs are indeed calculated using input from a coupled climate model. The basis for the ARTP is the regional climate sensitivities for four latitude bands, i.e., relationships between the pattern of RF caused by a certain forcing mechanism and the consequent surface temperature in a

given latitude region. These relationships are established from simulations with the GISS climate model and hence incorporate the response of the coupled climate system to the different forcing mechanisms and regions.

To fully account for the detailed regional temperature responses, one would naturally need a coupled climate model, for instance to capture potential changes in regional responses in a climate differing considerably from the present state. Currently, assessing the impact of individual sectors or mitigation measures is challenging and costly given the large natural variability of the global climate models, and requires significant resources and high technical skill. Emission metrics cannot replace the detailed information from climate models, but they serve as very useful tools to provide first-order estimates of the climate impact of various emissions, and they are based on information from complex models. Until recently, emission metrics have used globally averaged input to produced globally average impacts. Being able to include spatial information is an important step, in particular for the short-lived climate forcers that produce more heterogeneous RF than $CO_2$.

Although we believe we stated clearly the coupled-climate model origin of our ARTP values, at lines 198-200, to further emphasize this point, we have added the following in the introduction:
*"The ARTP uses a set of regional climate sensitivities to provide time-varying surface temperature response in four latitude bands to emissions, accounting for the regional RF caused by the emissions. These sensitivities are derived from simulations with a coupled climate model and express the relationship between the pattern of a radiative forcing and the consequent surface temperature in a given latitude band."*

Page 4, L110: I'm not sure you can do this with a CTM. The circulation doesn't adjust.
We realize that this sentence is unclear. As the reviewer points out, the emission metrics cannot be calculated directly from the combination of CTM simulations and radiative transfer modeling, but must include a representation of the climate (temperature) response to the given forcing. The CTM plus radiative transfer calculations allows us to capture detailed atmospheric chemistry and forcing. For emission metrics, the climate response is represented by an impulse response function. This is a simplified, global mean function, but is based on more complex climate models. In our analysis, we also use regional climate sensitivities which are based on coupled climate model simulations in order to capture information about the spatial distribution of the temperature response depending on the forcing mechanism and location.

To clarify, we have rephrased:
*"In this study we calculate GWP, GTP and ARTP for global and regional aviation emissions. We consider a broad set of forcing mechanisms and emissions in six separate source regions. Input of the aviation-induced radiative forcing of ozone and aerosols are obtained from simulations with the chemistry-transport model OsloCTM3 (Søvde et al., 2012) and radiative transfer calculations, while the radiative forcing from the formation contrail-cirrus is simulated with the ECHAM5-CCMod (Bock & Burkhardt, 2016b; a). Calculating global and regional emission metrics allows us to capture (…)"*

Page 5, L114: again: surface temperature response depends on the surface type. Is that accounted for?

The regional climate sensitivities that form the basis for the ARTP calculations are derived from simulations with a couple climate model, which includes a dependence on the surface type. See also response to the comment above.

Page 5, L117: how uncertain are the regional climate sensitivities?
This is a very good question. A full set of regional climate sensitivities (RCS) have so far only been estimated using one climate model by Shindell and Faluvegi (2009). Standard deviations over the 80 year model integrations show quite some variability in the response to certain species, at least in some regions. Two studies have repeated the experiments from Shindell and Faluvegi (2009) for BC with similar findings in terms of spatial distribution of forcing-response (Sand et al. 2013; Flanner 2013). The coefficients also seem fairly robust compared to the response to historical aerosol forcing in several other climate models (Shindell 2012). However, a formal quantification of their uncertainty is currently not possible and repeating the experiments to establish sensitivities with additional climate models is beyond the scope of this study, and would likely require a dedicated intercomparison exercise. Baker et al. (2015) showed that the zonal mean temperature response to BC emissions can differ significantly between different climate models both in magnitude and spatial distribution, while the models agreed better for $SO_2$ emissions. Some degree of model dependence in the RCSs cannot be excluded, in particular for some species.

To make the potential model dependence clear earlier in the manuscript, we have specified in the final paragraph of the introduction that the sensitivities have so far only been derived by one climate model. We have also incorporated some of the information above in the text.

Page 7, L173: I do not think you can throw out this term: it is a compensating effect or feedback. You could include the reduction in cirrus as a separate forcing, but now you are overstating the contrail impact.
We agree with the reviewer here and have included the 20% reduction in the RF of contrail-cirrus, assuming it is spatially constant, i.e., that the same impact occurs in each of our emission source regions.

Page 7, L184: What are these uncertainties?
We have clarified this sentence and added references to studies showing how there is a large range in aviation RF estimates reported in the existing literature. The sentence also fits better after the first two sentences in this paragraph and has been moved up. It now reads:
*"It should be noted that there is a broad range in the estimates of RF caused by the various aviation emissions reported in the literature (e.g., Brasseur et al. (2016); Lee et al. (2009)) and such uncertainties in RF will propagate to the emissions metrics."*

Page 7, L197: Does previous work show that this represents the physical response?
I have a hard time believing it works for non-uniform emissions in the vertical, or for aviation emissions.
The regional climate sensitivities are based on coupled simulations with the GISS model. One can argue whether current climate models represent the physical response, but they are at present the only tools we have.

As described later in the manuscript, the GISS simulations involve perturbing the RF resulting from anthropogenic emissions from all sources. Since, as the reviewer points out, the temperature response per unit RF can depend on the altitude of the RF for some species and regions, applying the sensitivities to situations which differ significantly from that originally used to derive them does introduce uncertainties. This is precisely why we take the analysis one step further to evaluate the application and point to where more research is needed in order to improve the general applicability of the ARTP.

Page 7, L200: Are these done for contrails? This is a cloud forcing, which is very different than other forcings. For example: contrail forcing is a SW and LW forcing that depends strongly on surface albedo. Do these sensitivities take account of that? If not, I'm not sure the method is valid.
We feel that the text starting at line 221 made clear what RTPs are used for the contrail-cirrus case, and the limitations of the assumptions we made.

The contrail-cirrus RF is quantified through detailed simulations with the ECHAM5-CCMod and takes the surface type and other cloudiness into account.

In terms of temperature response, we do not have regional sensitivities that have been derived from climate model simulations specifically for contrails (as we noted at line 221 to 223). Currently, such information does not exist in the literature, and we recognize that this introduces additional uncertainties in our analysis. To represent the regional response to a regional contrail-cirrus RF, we use the sensitivities based on the forcing-response relationship for $CO_2$ and sulfate. This includes trapping of long-wave radiation and reflection in the short-wave and the sulfate forcing likewise, as is well established, depends strongly on surface albedo. As described above, the sensitivities are based on coupled climate model simulations, which includes the dependence on surface type – for the $CO_2$ and sulfate forcing.

As we noted at line 226, the only two available studies of the climate response to contrails have shown that the global efficacy of line shaped contrails could be significantly lower than one, and the reasons for this may well be as a result of the distinct nature of the contrail forcing which the reviewer mentions. This lower surface temperature response per unit forcing compared to $CO_2$ could be accounted in our (and future) calculations by scaling the temperature response (as we discuss in Sect. 3.3). However, we do not have any information about how the efficacy could vary regionally, as we noted at line 229.

Page 8, L202: What does the statement in parentheses mean? What is a component? If the time is longer than the lifetime for a pulse emission, isn't the forcing gone?
This statement has been corrected, clarified and moved to the beginning of the next paragraph:
*"Equation 4 can be used for short-lived species where H is significantly longer than the lifetime of the species (typically days to weeks)."*

Page 8, L207: This presumes that all RF is created equal with respect to sensitivity. I think this is valid for GHG in the IR. I'm not sure if it is valid for SW effects, and probably not for clouds.

This is an important point. We already touch upon this issue in the following paragraph on efficacy, but we agree that a more detailed discussion would be valuable. Efficacies can be included in the application of metrics by adding a scaling factor, and will act to reduce or increase the temperature change to emissions. However, only a couple of studies have investigated the (global-mean) efficacy for aviation-induced forcing mechanisms and are limited to line shaped contrails and ozone, and we therefore include efficacy only as a point for discussion in Sect. 3.3.

The concept of efficacy, or climate sensitivity parameters, also relates to the RCS (as the reviewer touches upon in the next comment). These are in themselves a type of climate sensitivity parameters for the four forcing agents considered in the Shindell and Faluvegi (2009) study (BC, ozone, sulfate and $CO_2$), expressing differences arising from the forcing distribution in the horizontal.

We have therefore the discussion and added:
*"The temperature response per unit RF can differ between different forcing mechanisms, i.e., mechanisms can have their specific climate sensitivity parameter. This is often expressed as efficacy, defined as the ratio of the climate sensitivity parameter for a given forcing agent to that for specified changes in CO2 (Hansen et al., 2005). Efficacies can be included in the metric application as a scaling factor. However, presently only a few studies have investigated the efficacy of selected aviation-induced forcing mechanisms. The efficacy depends primarily on the spatial distribution of the RF, both in the horizontal and vertical. The $RCS_{i,l,m}$ implicitly include differences in efficacy of individual components arising from the horizontal forcing distribution (to the extent that the driving processes are accounted for the underlying climate model simulations). The $RCS_{i,l,m}$ are established for the four forcing agents BC, ozone, sulfate and CO2. Contrail-specific regional sensitivities do so far not exist. By using the average sensitivities of sulfate and CO2 to calculate the contrail ARTPs, we include both a longwave absorption and a shortwave scattering component. The efficacy of scattering aerosols and greenhouse gases is also likely less dependent on altitude than for absorbing aerosols. However, studies have indicated that the contrail efficacy may be as low as 0.3-0.6 (Ponater et al., 2005; Rap et al., 2010). Furthermore, little or no information about the dependence of the climate sensitivity parameter of contrails on the horizontal forcing distribution exist. It should also be noted that efficacies from the small sector-specific forcings can currently only be derived by scaling them to produce a forcing large enough to get a significant response in the model. This add an additional uncertainty to deriving reliable $RCS_{i,l,m}$. Using the $RCS_{i,l,m}$ for a different forcing agent to approximate the response to contrail-cirrus allows us to include a broader set of aviation-induced forcing mechanisms in our analysis, but is also an important caveat. As discussed in more detail in Sect. 3.4, the climate sensitivity parameter can also depend on altitude, in particular for absorbing aerosols and ozone in certain regions. We explore potential uncertainties in our analysis arising from such vertical differences by comparing the temperature responses estimated using the $RCS_{i,l,m}$ with temperature simulated by three additional climate models (see Sect. 2.3)."*

Page 9, L229: So you are trying to put these uncertainties into 'efficacy', which would affect the contrails by a factor of 2, but then ignore it? I don't think that is appropriate. Efficacy might be a way to address some of the issues I am referring to with contrails in this context.

The issue of efficacy is not ignored, but brought up again in Sect. 3.3, where we discuss how a lower contrail efficacy may affect the regional temperature response to present-day aviation emissions estimated with the ARTPs. However, currently, only two studies have estimated the efficacy of contrails and estimates of efficacy of other aviation-induced forcing mechanisms are very scarce (one study looked at aviation ozone). We emphasize again that these two studies provide information on the global-mean efficacy but we have no information about how this would influence regional climate sensitivities. Furthermore, in response to the question above, we have added a more detailed discussion about efficacy in general.

Page 9, L241: There are emissions scenarios for aviation that specifically address the non-uniform growth of emissions and the efficiency improvements. Why not use them? There are already RF estimates in the literature with these scenarios.

Here we try to relate RF to an emission and discuss whether to relate contrail RF to flown distance or to fuel use. Since fuel efficiency changes with time, the two metrics would give different results. Simulations for future air traffic do not help with the decision how to formulate the metric.

An assessment of the contrail formation and RF from regional aviation emissions under a different emission scenario and climate would of course be an interesting study on its own. However, repeating the regional model runs with a different emission inventory requires resources beyond what we have available. Our global emission metric calculations could be repeated using input of RF from studies that the reviewer alludes to and could make an interesting sensitivity study. However, since we focus primarily on the regional metrics here, we leave this to further applications of our framework.

Page 10, L270: I do not think the climate system is linear enough to small perturbations to make this a valid assumption. Can you provide a reference demonstrating that multiplying a forcing by an order of magnitude or more yields a linear scaling to the response to the small perturbation in a coupled climate model? I see now there is a reference below. Is it valid?

Here, the reviewer seems to questions the approach and requests a citation, but then (without any justification) questions the validity of our citation upon discovering that we already have one.

Applying a scaling factor (or performing very long model integrations) is a common approach that makes it possible to use climate models to study the response to specific sectors or perturbations. Because of their small size, the use of unscaled forcings to derive reliable responses is very challenging and costly due to the climate model's internal variability.

The Shine et al. (2012) study illustrates that non-linearities may indeed arise from using different scaling factors in the specific case of BC perturbations from road transport. In contrast, the experiments carried out by Mahajan et al. (2013) (BC x 1, 2, 5 and 10 in the CAM4) actually show a quite linear relationship between the magnitude of the BC perturbation and global temperature response. Nonetheless, this is why we encourage the reader to keep the potential

uncertainties in mind when interpreting the absolute magnitudes of the simulated temperature response (see response to next comment).

Page 10, L274: But isn't the temperature response the heart of this paper? Yet it is to be used 'with care'? I think you might need to quantify the potential uncertainties here.
At the core of the evaluation of our metrics against the additional climate model simulations is the geographical pattern of temperature response. The sentence in question aims to caution the reader against potential uncertainties in *absolute magnitude* that may arise from applying a scaling factor in the climate model (as the reviewer also points out in the comment above). A quantification of uncertainties arising from the use of different magnitude scaling factors require a large set of additional model simulations and is beyond the scope of the study. However, we agree that this can be made clearer and have rephrased:

*"(…) and potential uncertainties should be kept in mind when considering the absolute magnitude of temperature responses."*

Page 10, L278: I do not think figure S1 is correct. I do not see how a 2ppb ozone perturbation causes a 1K surface temperature change in Figure S1. Please clarify or justify that.
Fig. S1A shows the ozone perturbation caused by present-day aviation NOx emissions simulated by the OsloCTM3 before scaling for input to the CESM. We thank the reviewer for noticing that this was not specified in the figure caption. It has now been added:
*"Figure SI 1: (A) Annual mean ozone concentration change from OsloCTM3 caused by global aviation NOx emissions and (B) annual mean surface temperature response to the aviation ozone perturbation scaled by a factor 40 as simulated by CESM1.2. Hatching indicates statistical significance at the 0.05 level."*

The text in the manuscript has also been corrected accordingly:
*"Figure SI 1A) shows the zonal annual mean ozone concentration change caused by global aviation NOx emissions from the OsloCTM3 (i.e., before scaling), while Fig. SI 1B) shows the annual mean CESM2.1 temperature response to the scaled ozone perturbation (hatching indicates statistical significance at the 0.05 level)."*

Page 11, L287: But I thought above the forcing was an impulse, I.e. Non- constant?
Yes, the emission metrics ARTP(H) are pulse-based. As described in Shindell (2012), the RCSs can also be used to estimate the approximate equilibrium temperature response to a constant forcing by multiplying the sum of RCS-weighted regional RFs by the equilibrium climate sensitivity (in our study the ECS inherent in the IRF from Boucher and Reddy (2008)). We have modified the text and hope this makes it clearer:
*"For comparison with climate model results, we use the regional climate sensitivities to estimate the regional equilibrium temperature response ($\Delta T_{i,r,m}$) to a constant forcing following Eq. 6 of Shindell (2012)*

$$\Delta T_{i,r,m} = \sum_l RF_{l,r} \cdot RCS_{i,l,m} \cdot ECS \tag{8}$$

*and adopting the equilibrium climate sensitivity (ECS) inherent in the IRF from Boucher and Reddy (2008) described above."*

Page 11, L294: This is confusing regarding emission units: since they are rationed to CO2, shouldn't they be relative to aviation CO2, then the comparison with aviation CO2 is explicit. Pardon me if that is not the appropriate definition, but I'm not sure how to interpret the very large BC GWP number.

We adopt the standard method of reporting GWPs as being relative to $CO_2$ and since the radiative efficiency of $CO_2$ (which goes into the emission metric calculations) is the same for all sources of $CO_2$ the reviewer's comment is not relevant. The unitless GWP and GTP of individual species is the AGWP and AGTP normalized by the $AGWP_{CO2}$ and $AGTP_{CO2}$. However, for $CO_2$, the radiative efficiency is the same regardless of the source. Of course, if one wants to for instance estimate the temperature response to aviation $CO_2$ using the $AGTP_{CO2}$, one needs to multiply by the $CO_2$ emissions from this sector.

Page 11, L305: But does this treat compensating LW and SW effects which are a function of surface type?

Yes it does.

Page 12, L323: Dry conditions in S. Asia? Really? In some seasons. What about the summer monsoon?

We agree with the reviewer that this may be a confusing sentence to some readers. We are referring to the high altitudes where the majority of aviation emissions occur in this region, as is noted in the first part of the sentence in question. However, drier conditions at these altitudes are not limited to this region. Furthermore, this region covers both parts of the Middle East and Arabian Peninsula, in addition to India. We have rephrased for clarification:

*"(…) dominated by emissions at high altitudes (i.e., few flights landing or departing within the region), where conditions are drier (i.e., less wet scavenging of the aerosols)."*

Page 13, L345: I'm not sure I agree with this. How do you deal with non local energy balances in each region?

As we noted above, the regional climate sensitivities are based on coupled climate model simulations where the perturbation is imposed separately in the four individual latitude band and the consequent temperature response averaged over each of these. Hence, these non-local energy budget terms are included.

Page 13, L364: Is it valid to apply RCS from one model to CTM results from another model? They might be vastly inconsistent, particularly in remote regions.

There are of course differences between the different chemistry-transport schemes, such as in vertical profiles of trace gases and particles or polar transport. However, we are not aware of literature that support that current models are vastly inconsistent. The reviewer's contention is not supported by any references and seems speculative.

Page 13, L370: I'm not sure the remote effects being larger make physical sense. See next comment.

Our results are supported by the existing literature. BC aerosols from aviation that are emitted in the 60-90°N region, or transported in there from lower latitudes, are located at high altitudes. Several studies have found that the efficacy of BC decreases with altitude and that in the Arctic, high altitude BC even causes a surface cooling (Flanner 2013; Ban-Weiss et al. 2011; Samset

and Myhre 2015). In the case of aviation, the contribution to temperature response in the 60-90°N region caused by within-region BC is cooling. In contrast, BC at latitudes further south warms the atmosphere locally, which in turn gives a warming impact on the Arctic transport of energy, not transport of aerosols. Since the Shindell and Faluvegi (2009), two other studies have found the same feature (Flanner 2013; Sand et al. 2013).

Studies have also shown that the relative importance of local and remote BC for the Arctic temperature response dependence on the source and/or region: BC located at lower latitudes in the Arctic have a warming impact on surface temperatures. Hence, for emissions from sectors/regions closer to or in the Arctic, the local contribution is more important (e.g., Lund et al. 2014).

While the scientific community's understanding of the climate response to BC forcing may not yet be complete, we are not aware of published literature that disagrees with the findings of the above mentioned studies.

Page 14, L378: Here it looks like the GISS results are convolving the effect of transport with the effect of aerosols: they co-vary, but the aerosols do not cause warming. They just come in with warmer air. I think this highlights some of the problems with this methodology. I think you draw the wrong conclusion here.
If we understand this comment correctly, we think this is precisely the point behind what we (and previous analyses) argue: The remote impact is not caused by the direct warming by the aerosols, but by the transport of the air that has been warmed. See also response to comment above.

Page 14, L390: Unless emissions occur in the stratosphere, in which case contrail formation drops.
Yes, good point. We have specified this in the text.

Page 15, L406: what about significant changes in meteorology: warming reduces contrails, and also changes the tropopause height.
Yes, this is a relevant point, but it is one we addressed through the citation in the very next sentence. To clarify further we elaborate:
*"Furthermore, future changes in climate may alter the meteorological and dynamical conditions, and hence affect the potential for contrail-cirrus formation in a given region (Irvine and Shine, 2015)"*
We also note that the reviewer is being too simplistic – the change in tropopause height leads to a moistening of the air at a given altitude in the extratropics, and likely leads to more contrails. This point is too detailed for this paper, but our sentence deliberately refers to changes rather than specifying signs.

Page 16, L454: I'm not sure exactly what is being plotted in figure 3: please explain further and in a more detailed caption. Temperature change in each region by different emissions in each region? Are the asterix just a label for the bars or does their vertical position mean anything.
The following has been added to the caption:

*"Figure 3: Regional and global mean temperature change by species and source region after A) 20 years and B) 100 years following a one-year pulse of emission from the present-day aviation sector in each source region. The asterisk shows the net temperature response in the respective latitude band for each emission source region, while the bars show the contribution from each species to the net."*

And the text has been slightly modified for clarification:
*"Figure 3 shows the temperature change (net and contribution from each species) in each latitude band 20 and 100 years after a one-year pulse of present-day aviation emissions in each source region."*

Page 17, L484: If previous studies only did surface emissions (e.g. Stohl et al 2015 and Shindell 2012), then how will some of the terms from these models be valid (e.g. GISS study with RCS I think it is)?
The Stohl et al. 2015 study used the same approach as in our analysis and calculated the ARTPs for a set of components, emission source regions and seasons. They did not derive additional, new regional climate sensitivities, but compared the ARTPs with results from transient climate model runs (similar to our approach, but we consider the simulated equilibrium temperature responses). However, their emission source regions included all anthropogenic sources, i.e., mainly surface sources, and it is unclear if the conclusions from their evaluation of the ARTP can be extrapolated to the aviation sector.

Similarly, Shindell (2012) did not derive new RCS, but used those from the Shindell and Faluvegi (2009) paper and evaluated their robustness compared to historical aerosol forcing simulated by several other models. Again, total anthropogenic aerosol forcing is largely determined by emissions from surface sources, with only a small contribution from aviation.

Hence, while the application of the RCS have been evaluated in previous studies, our analysis is novel because it considers a specific sector. Neither of the abovementioned studies derived new RCS or include new results that could be directly incorporated in the current study. Furthermore, discussing the potential uncertainties in their analyses or the validity of the results is beyond the scope of our study.

Page 18, L511: I could also argue using this figure that the ARTP concept only works within the main emissions region, and is off by 50% or more relative to a physical calculation.
There are of course several ways these findings could be phrased and we could arguably explicitly say that the 28°S-28°N and 28-60°N latitude bands, where we find the best agreement, are the main emission regions. Hence, we feel that this has already been covered in the previous sections of the paper. However, since the ARTP concept is based on coupled climate model simulations, we do not agree with the interpretation in the last part of the reviewer's argument. And that final part aside, the rest of the argument is just a different way to phrase the same findings.

Page 19, L528: I think this whole analysis highlights that the ARTP concept is very model dependent and not a general concept because of strong dependencies you have identified here.

We think it is important to separate the concept from the input data here. While the ARTP is a useful tool, and is the only emission metric to provide information at a sub-global scale, there is a certainly a need for additional studies to investigate the robustness of the regional climate sensitives. It is also a relatively new concept, and presently we do not know how model dependent the sensitivities are. As discussed in response to one of the comments above, they seem to seem fairly robust compared to the response to historical aerosol forcing. However, to further examine the model dependence it is necessary to repeat the original simulations with additional climate models, which is of course a major undertaking. What we have shown in our analysis is that there are indeed uncertainties, as well as a need for improved knowledge of vertical dependence in the forcing-response relationships, not only globally, but also in different regions. Our study presents a framework that can expanded and improved in the future when/if new, more robust and detailed estimates of regional climate sensitivities becomes available.

The ARTP has already been used in many assessments and we believe analyses such as ours are important to highlight the potential applications and associated uncertainties and point to where future studies could contribute. We also conclude that further work is needed for a more robust and general application of the concept, in agreement with the reviewer's argument.

Page 19, L546: I think you are going to have the same problem with contrails, which also have strong vertical effects, and regional forcing dependent on the climate system itself.
We agree that there may be some strong vertical effects for contrails, but at this stage this can only be speculation – unlike the case for BC where a number of pre-cursor papers already demonstrate the effect. Also the situation is quite different for BC and contrails. There is only a limited vertical domain over which contrails can physically exist, whereas there is no similar physical constraint on where the BC can be.

Page 19, L553: Can you do an uncertainty analysis here? Where are the largest uncertainties and model dependencies in the analysis.
We agree with the reviewer that a better quantification of the uncertainties is valuable. Uncertainties in ARTP arise from a number of factors, including emissions, RF and climate sensitivity. Currently, there is insufficient information to quantify the contribution from the regional climate sensitivities. Similarly, uncertainties in the aviation emission inventory are not available. We have however included an analysis of the impact of uncertainties in the different RF mechanisms and the global climate sensitivity on the global mean temperature response to the regional aviation emissions. We have also expanded the discussion on efficacy and aerosol indirect effects.

Page 21, L589: Where do you demonstrate that the response is stronger than the forcing? And how does this analysis include effects of the large scale circulation and feedbacks? You are breaking most of those feedbacks with use of a CTM and unitless response functions from another model.
This can be seen by comparing the latitudinal distribution of RF values given in Table SI2 with the distribution of the ARTPs (although the absolute magnitudes are not directly comparable since the ARTPs are given per unit emissions and as a function of time). Since we do not present the RF in separate figures and additional calculations are needed for a direct comparison of absolute values, we agree that this should be made clearer and have added (in Sect. 3.2):

*"This can be seen by comparing the latitudinal distribution of the RF values given in Table SI2 with that of the ARTPs (note that absolute magnitudes are not directly comparable since the ARTPs are given per unit emissions and as a function of time)."*

The effects of large-scale circulation and feedbacks are included through the use of the regional climate sensitivities (forcing-temperature response relationships), which are derived from coupled climate model simulations. See also response to previous comments.

Page 21, L606: Add to this some scaling by a factor of 40 for emissions in here somewhere, and I think this is problematic.
To clarify, it is not the emissions that are scaled by a factor 40, but the concentrations. Furthermore, this is not done in the ARTP calculations, but in the climate model simulation used for evaluation (since otherwise there is a difficulty in detecting the signal above the model-generated variability).

Page 21, L608: At least I agree with the negative conclusion here: it is not clear to me that this concept has a lot of utility for aviation. Especially since you have left out aerosol indirect effects. Anything that involves clousds I think is highly problematic using this method. Would like to see a bit more uncertainty analysis to quantify where the uncertainty lies.
Concerning the first sentence of the reviewer's comments, we note again that it is very important to separate the concept from the characteristics of the input data we have had to use here. We believe that the concept has great utility and hopefully studies like ours will stimulate the development of more "bespoke" input data.

An important application of emissions metrics is the assessment and comparison of the impact of various components under different emissions scenarios or of emission changes following mitigation measures. This in turn provides a basis for comparing and evaluating different mitigation strategies or policies. For instance, if a mitigation measure has different effects on NOx, $CO_2$ or contrails, the subsequent climate impact of these species over time can be compared using our metrics. So while we agree that our frameworks has limitations when it comes to for calculating the total climate impact of aviation since we do not include indirect aerosol effects, we do not agree that there is no utility for the aviation industry. Furthermore, once established, our framework can be expanded and improved as science progresses and the knowledge of the aviation-induced forcing mechanisms is improved.

An uncertainty analysis has been added - see response to comment Page 19, L553 above.

Page 28, L882: What are the units of BC? Per unit of fuel or something else? Not clear from the caption.
BC, OC and $SO_2$ are per unit BC, OC and $SO_2$, respectively. This has been specified in the captions of both Table 1 and 2.

Page 32, L922: What are the units here? Mili-kelvin? Over what area? Do the regional colors (asterisk) correspond to the emissions colors? I'm not sure what is being plotted here.

The caption has been modified to include a more detailed description of symbols and colors (see also response to comment above). The unit (milli kelvin) is given on the y-axis and the region/area is given on the x-axis (and now specified in the caption).

---

## Author Comment (AC2)

**Response to review of *"Emission metrics for quantifying regional climate impacts of aviation"* by Marianne T. Lund, Borgar Aamaas, Terje Berntsen, Lisa Bock, Ulrike Burkhardt, Jan S. Fuglestvedt and Keith P. Shine**

We thank the reviewer for the careful review and useful comments and suggestions. Responses to individual comments are given below.

**Anonymous Referee #2**

Major Comments:
142: The RF kernels of Samset and Myhre (2011) are valuable because they are vertically distributed. But they are also limited in their spatial coverage. How do the authors map between the regions in their study and those in Samset and Myhre (2011)? For example, it would seem the latter does not provide any results for the author's SAS and SPO regions. A more complete 2D spatial mapping of aerosol direct radiative forcing efficiencies is provided in Henze et al. (ES&T, dx.doi.org/10.1021/es301993s, 2012). Perhaps results from these two studies could be combined to provide a more complete analysis? Or at least findings from the latter could be used to provide some sense of the uncertainty involved in using only the Samset and Myhre regions as the basis for the present work.
The RF kernels are calculated by applying globally uniform aerosol perturbations in each vertical layer, thereby providing full 3D fields (as already mentioned in the text), not kernels for separate geographical regions (Samset and Myhre 2011 then averages the RF globally and in different regions to illustrate geographical differences). We realize that this may not be clear from the cited literature and have added the following clarification to the methodology section in the current study:

*"(..) where the radiative forcing per burden was derived by imposing globally uniform perturbations of given aerosol species at 20 different pressure levels from the surface to 20 hPa."*

A comparison of results derived by using the RF kernels and sensitivities from the adjoint modeling of Henze et al. (2012) (as well RF from full radiative transfer modeling) would be an interesting sensitivity study as in both cases RF values depend on the chemistry-transport model used to establish the background aerosol concentrations (OsloCTM2 and GEOS-chem). However, we feel that this is outside the scope of our study.

180-185: This argument feels a bit thin, given that some aspects of aerosol cloud interactions are at least better known that others. At the very least, could uncertainties owing to these processes be carried through the calculation, so that we know when uncertainties in these effect may alter the sign of the next outcome?
While we agree that the understanding of interactions between anthropogenic aerosols and liquid clouds has improved, we still maintain that the impacts of aerosols on ice clouds remain highly uncertain. Furthermore, estimates of sector specific aerosol-cloud RF remain scarce. Studies of the potential effects of aviation BC on large scale cirrus clouds have yet to agree even on the sign the radiative forcing, and the magnitude of the impact depend heavily on assumptions in the

models, ranging from -350 mW/m$^2$ to + 90 mW/m$^2$ even in a single study (Zhou and Penner 2014). Of course, if either of these number represent the actual impact, this effect would dominate the climate effect of aviation.

To our knowledge, only three studies have presented estimates of the impact of global aviation aerosols on liquid clouds, with results ranging from -46 to -15 mW/m2. We do not have the resources to quantify aerosol-cloud interactions of regional aviation emissions, but these three studies at least agree on the negative sign the global mean forcing, which could offset a considerable fraction of the warming from other components in the short term. So we have rewritten and added more detail:

*"Moreover, our results do not include effects of aerosol-cloud interactions, which is an important caveat. Studies have suggested a potential impact of aviation BC on large scale cirrus clouds, but have yet to agree even on the sign of the radiative forcing (Zhou & Penner, 2014). A few studies have investigated effects of aviation emissions on liquid clouds, with global mean RF estimates ranging from -46 to -15 mW/m$^2$ (Gettelman & Chen, 2013; Kapadia et al., 2016; Righi et al., 2016), i.e., a cooling that could offset a considerable fraction of the positive RF of contrail-cirrus and ozone on a global scale. However, at present uncertainties in these estimates are also very large, and we consider that their inclusion here would be premature."*

For Fig 4: why compare ARTP(20) and ARTP(100), when a more direct and fundamental comparison would be to just consider the RCS's? The RCS is what other people will need, if they are to use the results from this study themselves to calculate ARTPs. At the very least it would be quite useful to compare the RCS values in addition to the existing figures using ARTP in particular years.

While it is correct that the RCSs are needed as input if other people are to repeat the ARTP calculations, for instance with updated RF estimates, the ARTPs are what is needed in order to be able to make first-order estimates of the regional temperature impact of given emissions (the core application of the emission metrics). The RCS also do not have a temporal resolution, but are constant factors to distribute the impacts regionally. So, for instance the behavior of the net NOx impact over time would not be illustrated by the RCSs.

Furthermore, we have not estimated new RCSs in this study (Fig. 4 shows a comparison against temperature response to global aviation – not the response to perturbations in individual latitude bands – and only for NOx). The RCSs have been tabulated in previous literature, but to further aid the reader, we have added a summary table in the supplementary material.

General: For other people to make use of these results, it is useful to provide more information on the aviation emissions used in this study. The authors should provide a table of emissions by species and region, and they should provide separate total for emission by takeoff vs cruise altitudes. While it would be great if they could provide metrics broken down by the later category as well, I'd guess that would involve repeating a lot of calculations. But at least providing the details of the inventory they used would allow future users to be able to scale evaluations of the climate impacts of their own inventories accordingly, given some knowledge of how the authors' inventory was distributed vertically.

A table with total aviation emissions by species and region is already included in the supplementary material.

To separate metrics by cruise and landing/takeoff (LTO) operations we would indeed need to repeat our model simulations, which require additional resources and time that are not available. Guidance on how to access and use the AEDT emissions in atmospheric models is provided in a technical note by Barrett et al. (2010), including how to define and estimate emissions during LTO, allowing users to for instance use our metrics with emissions broken down by category. We have added the following:
*"Guidance on how to access and use the AEDT emissions in atmospheric models is provided by Barrett et al. (2010). For input to the OsloCTM3, emissions are interpolated to the model's horizontal and vertical resolution, and averaged monthly."*

Minor comments:
35-38: True, but this is evident from the fact that the RCS's in e.g. Shindell 2012 are not uniform. So it is a bit odd to place this in the abstract, although I agree the application does being attention to the issue.
The lack of one-to-one relationship between regional forcing and temperature response is one of the key features that can be emphasized by the use of sub-global, temperature-based emission metric such as the ARTP. It also points to added value of moving beyond RF-based emission metrics, such as the GWP. While this may be clear to the scientific community, it is not necessarily obvious to decision makers. We therefore feel that is an important point to highlight.

40-41: The feels a bit obvious (biggest emissions have the biggest impact) ăˇAˇT would discussing the impact per emission be of more interest?
We do not feel that it is obvious that the bigger emissions lead to a largest net warming impact "in all latitude bands" (which is what we write in the abstract) as that is a result that emerges from our analysis. Since the reviewer flags this as a minor comment, we prefer to keep this as it is. To summarise the relationship between each emitting region and each response region (as shown in Figure 2) in a short sentence in the abstract would be very challenging.

66: This statement is missing references.
There are several studies of how regional emissions affect atmosphere and climate; we have added to examples – one general and one aviation specific: Berntsen et al. (2005, doi: 10.1007/s10584-006-0433-4) and Stevenson and Derwent (2009, doi: 10.1029/2009gl039422).

78: The phrase "in a grid cell" is vague (we don't know yet how big your model grid cells are) and also ambiguous with regards to whether you are referring to grid-scale changes in temperature or grid-scale changes in emissions.
The wording on line 78 is in fact "at a grid point level", which is meant to be general so as not to make the link to a specific model or resolution. The sentence also states "temperature response and other climate variables", thereby not referring to emissions. However, to clarify have rephrased to:
*"(…) very detailed spatial scales (e.g., grid point level)"*

83-84: Can the authors reference any in particular?
Specific measures to reduce emissions are implemented at the sectoral (as well as regional/local) level. To assess their effectiveness in terms of reducing the climate impact, one needs to know how the sector contributes to climate change to begin with. We feel that this is a very generic statement that does not require specific references.

91-93: How much uncertainty / error can this aggregation lead to?
This is not easily quantified in a general way, as it depends on, among other things, which measure, which impact and which driver is considered. For instance, in Shine et al. (2005) it was found that the net temperature response to NOx emissions was positive in the Northern Hemisphere and negative in the Southern (due to different relative importance of ozone production and methane reduction), resulting in a very small on global-mean temperature impact. Lund et al. (2012) found a factor 2-7 higher impact of aviation NOx when assuming a non-linear impact function compared with one based on global global-mean input. In terms of RF, Burkhardt and Kärcher (2011) estimated a global-mean contrail RF of 37 mW/m2, but regionally values were up to 300 mW/m2.

105-110: See also Sand et al., Nature Climate, doi:10.1038/nclimate2880, 2015.
Yes, that is a relevant reference and has been included.

183: I don't understand what mechanisms this refers to. Please be more specific and provide references.
This refers to the existing uncertainties also for contrail-cirrus, NOx and aerosol effects considered in our study. We agree that the current wording should be improved. The sentence also fits better after the first two sentences in this paragraph and now reads:
*"It should be noted that there is a broad range in the estimates of RF caused by the various aviation emissions reported in the literature (e.g., Brasseur et al. (2016); Lee et al. (2009)) and such uncertainties in RF will propagate to the emissions metrics."*

Table 1: could you list NOx in the first half at the bottom the list of species, so that it is easier to compare these numbers to the results for NOx from other studies listed below?
Good point. We have changed the order (and for consistency, also in Table 2).

Table 1: I must be missing something ã˘A˘T the GTP and GWP metrics are computed by emitted species (i.e., SO2 instead of sulfate), yet the authors report values for nitrate(Table 1), and these are reported separately from NOx emissions, even though nitrate is formed secondarily in the atmospheric from NOx. Can the authors please explain this more?
We thank the reviewer for pointing this out, the tables are not labeled correctly. The NOx entry includes the impact of NOx on ozone and methane, while the nitrate label is the effect of NOx-induced nitrate formation. To make this clear, the table should read NOx-nitrate and NOx-ozone-methane. We separated out the nitrate contribution since to our knowledge no previous estimates exist, making it difficult to compare the NOx metrics with previous literature if it was included. However, we realize that this may be unclear in further application of the metrics, since these two NOx values would first have to be added. So, we have combined to one metric for the net NOx effect. Tables 1 and 2 have been changed and the text clarified:

*"Our estimates also includes the cooling effect from NOx-induced formation of nitrate aerosols, which has to our knowledge not been accounted for in any previous GWP and GTP estimates."*

Fig 2: Please explain the difference between the color bars vs star points in panels E and F, and define O3 vs O3PM in the figure caption itself.
This has been included in the caption.

381: Also Lacey et al. (PNAS, doi:10.1073/pnas.1612430114, 2017) used ARTPs to investigate this for cookstove emissions.
This new study of cook stove emission is very interesting, but as far as we understand it focuses on the impact of regional emissions on global temporal temperature change, determined using regional climate sensitivities, but not explicitly presenting results to support the current sentence.

132: Is this perturbation positive or negative? Does it make a difference, for SO2 and NOx?
The perturbation is negative, i.e., we remove 20% of emissions. However, the difference between the reference and perturbed run is chosen so as to determine the impact of the aviation sector emissions (rather than the impact of a specific emission increase or decrease), so in that sense it will not matter. It is possible that non-linearities in the chemistry would result in differences compared to a positive perturbation. The impact of such non-linearities when applying different size perturbations with the same magnitude has been found to be relatively small (e.g., Hoor et al. 2009; Lund et al. 2014; Myhre et al. 2011).

Fig 2: Given the factor of 0.5 in Eq 2, why isn't O3PM 50% that of the CH4 in panels E and F? Is this a consequence of the spatial re-scaling from Fry 2012? If so, I would have expected it to be less than 50% in some regions and greater than 50% in others.
This is because $CH_4$ includes the RF of the methane-induced stratospheric water vapor change as described in Sect. 2, while the RF $O_3$PM is estimated as 0.5 of the "pure" RF $CH_4$. We have added a clarification in the figure caption. We also discovered a very small error in a couple of the O3PM RF. These have been corrected, but do not affect our results.

222: Why not use the RCS for sulfate for sulfate, rather than for the mean of CO2 and sulfate? What RCS is used for nitrate (although not clear how nitrate is treated anyways)?
The same RCS is applied for nitrate as for sulfate and OC. This has been clarified in the text. We follow the approach of previous studies when adopting the mean sulfate/CO2 RCSs (Collins et al. 2013; Shindell and Faluvgi 2010). There has been one application with the sulfate-only RCS as well (Shindell 2012). In fact, when comparing the RCS, they are very similar (Shindell 2012 and Shindell and Faluvegi 2010), and choosing one over the other are not likely to significantly affect our findings.

492-497: This statement is a convolution of two issues that could be separated, which are that the RF of O3 per ppb is horizontally and vertically variable, and that the climate response to this RF is also variable.
A good point. We have added:
*"(…) which in turn also depends on altitude (e.g., Olsen et al. (2013))."*

536-538 and 550-551: Fig 4 only shows the normalized results, so it is hard to know how much of an overestimate the authors are talking about here. Can they also provide the absolute results? The lines in question discuss the possible overestimation in more general terms, based on the existing literature on BC forcing-response. We do not currently have sufficient information on the vertical sensitivity in the BC forcing-response in latitude bands other than the Arctic to say how large such an overestimation could be. The regional distribution in Fig. 4B gives an indication, but it is only one climate model, and given the uncertainty in the magnitude in temperature response to BC in current global climate model we have some reservations against discussing the absolute magnitudes. This is further strengthened by slight differences in the experimental setup and input data in the HadSM3/ECHAM versus CESM/RTP, which, as described in the text, means that differences in absolute magnitude cannot be entirely attributed to model differences. All in all and given the scope of our study, we feel that a focus on geographical distributions is preferable.

Did the authors consider using ARTPs for the land-only response from Shindell 2012? The reviewer brings up an interesting suggestion. A comparison of the response estimated with the land-only regional climate sensitivities against the corresponding temperature response simulated by other climate models could be an interesting part of a more detailed evaluation/sensitivity study. However, here we are limited by the availability of output from the HadSM3 and ECHAM models.

Technical corrections:
87: as a bridge
133: sulfer dioxide
137: each region are
303: in the present analysis we
Corrected.